# Bidirectional synaptic plasticity rapidly modifies hippocampal representations

Aaron D Milstein[1,2], Yiding Li[3], Katie C Bittner[4], Christine Grienberger[3], Ivan Soltesz[1], Jeffrey C Magee[3]*, Sandro Romani[4]*

[1]Department of Neurosurgery and Stanford Neurosciences Institute, Stanford University School of Medicine, Stanford, United States; [2]Department of Neuroscience and Cell Biology, Robert Wood Johnson Medical School and Center for Advanced Biotechnology and Medicine, Rutgers University, Piscataway, United States; [3]Howard Hughes Medical Institute, Baylor College of Medicine, Houston, United States; [4]Howard Hughes Medical Institute, Janelia Research Campus, Ashburn, United States

**Abstract** Learning requires neural adaptations thought to be mediated by activity-dependent synaptic plasticity. A relatively non-standard form of synaptic plasticity driven by dendritic calcium spikes, or plateau potentials, has been reported to underlie place field formation in rodent hippocampal CA1 neurons. Here, we found that this behavioral timescale synaptic plasticity (BTSP) can also reshape existing place fields via bidirectional synaptic weight changes that depend on the temporal proximity of plateau potentials to pre-existing place fields. When evoked near an existing place field, plateau potentials induced less synaptic potentiation and more depression, suggesting BTSP might depend inversely on postsynaptic activation. However, manipulations of place cell membrane potential and computational modeling indicated that this anti-correlation actually results from a dependence on current synaptic weight such that weak inputs potentiate and strong inputs depress. A network model implementing this bidirectional synaptic learning rule suggested that BTSP enables population activity, rather than pairwise neuronal correlations, to drive neural adaptations to experience.

*For correspondence:
jcmagee@bcm.edu (JCM);
romanis@janelia.hhmi.org (SR)

Competing interest: The authors declare that no competing interests exist.

## Editor's evaluation

This manuscript uses a combination of high-quality in vivo electrophysiology and modelling to demonstrate that Behavioural Time Scale Plasticity (BTSP) is bidirectional, and the amplitude and direction of this plasticity are dictated by the current weight of the inputs and not by the correlated activity of pairs of neurons. These findings challenge our current views on synaptic plasticity, which are primarily based on Hebb's concept. In addition, the network model used in this study demonstrates that this type of plasticity can rapidly reshape population activity to respond to environmental clues. This study will be of interest to the broad neuroscience audience and foster new ideas on biological and artificial learning.

## Introduction

Activity-dependent changes in synaptic strength can flexibly alter the selectivity of neuronal firing, providing a cellular substrate for learning and memory. In the hippocampus, synaptic plasticity plays an important role in various forms of spatial and episodic learning and memory (*Nakazawa et al., 2004*). The spatial firing rates of hippocampal place cells have been shown to be modified by experience and by changes in environmental context or the locations of salient features (*O'Keefe and Conway, 1978*; *Mehta et al., 1997*; *Lever et al., 2002*; *Dupret et al., 2010*; *Zaremba et al., 2017*;

**eLife digest** A new housing development in a familiar neighborhood, a wrong turn that ends up lengthening a Sunday stroll: our internal representation of the world requires constant updating, and we need to be able to associate events separated by long intervals of time to finetune future outcome. This often requires neural connections to be altered.

A brain region known as the hippocampus is involved in building and maintaining a map of our environment. However, signals from other brain areas can activate silent neurons in the hippocampus when the body is in a specific location by triggering cellular events called dendritic calcium spikes.

Milstein et al. explored whether dendritic calcium spikes in the hippocampus could also help the brain to update its map of the world by enabling neurons to stop being active at one location and to start responding at a new position. Experiments in mice showed that calcium spikes could change which features of the environment individual neurons respond to by strengthening or weaking connections between specific cells. Crucially, this mechanism allowed neurons to associate event sequences that unfold over a longer timescale that was more relevant to the ones encountered in day-to-day life.

A computational model was then put together, and it demonstrated that dendritic calcium spikes in the hippocampus could enable the brain to make better spatial decisions in future. Indeed, these spikes are driven by inputs from brain regions involved in complex cognitive processes, potentially enabling the delayed outcomes of navigational choices to guide changes in the activity and wiring of neurons. Overall, the work by Milstein et al. advances the understanding of learning and memory in the brain and may inform the design of better systems for artificial learning.

---

*Turi et al., 2019*; *Ziv et al., 2013*; *Muller and Kubie, 1987*; *Bostock et al., 1991*; *Fyhn et al., 2007*; *Leutgeb et al., 2005*). These modifications can occur rapidly, even within a single trial (*Hill, 1978*; *Mehta, 2015*; *Monaco et al., 2014*; *Bittner et al., 2015*; *Bittner et al., 2017*; *Diamantaki et al., 2018*; *Jezek et al., 2011*; *Geiller et al., 2017*; *Bourboulou et al., 2019*; *Zhao et al., 2020*). Here, we investigate the synaptic plasticity mechanisms underlying such rapid changes in the spatial selectivity of hippocampal place cells.

Various forms of Hebbian synaptic plasticity have been considered for decades to be the main, or even only, synaptic plasticity mechanisms present within most brain regions of a number of species (*Magee and Grienberger, 2020*). The core feature of such plasticity mechanisms is that they are autonomously driven by repeated synchronous activity between synaptically connected neurons, which results in either increases or decreases in synaptic strength depending on the exact temporal coincidence (*Gerstner et al., 2018*; *Keck et al., 2017*; *Shouval et al., 2010*; *Song et al., 2000*). This includes the so-called 'three-factor' plasticity rules that, in addition to pre- and postsynaptic activity, depend on a third factor that extends the time course over which plasticity can function (*Magee and Grienberger, 2020*; *Gerstner et al., 2018*; *He et al., 2015*; *Yagishita et al., 2014*). To implement these three-factor plasticity rules, it has been proposed that correlated pre- and postsynaptic activity drives the formation of a synaptic flag or eligibility trace (ET) that is then converted into changes in synaptic weights by the delayed third factor, usually a neuromodulatory signal (*Gerstner et al., 2018*; *Sajikumar and Frey, 2004*; *Frey and Morris, 1997*).

Recently, we reported a potent, rapid form of synaptic plasticity in hippocampal CA1 pyramidal neurons that enables a de novo place field to be generated in a single trial following a dendritic calcium spike (also called a plateau potential) (*Bittner et al., 2015*; *Bittner et al., 2017*; *Diamantaki et al., 2018*). This form of synaptic plasticity, termed behavioral timescale synaptic plasticity (BTSP), rapidly modifies synaptic inputs active within a seconds-long time window around the plateau potential. This relatively long time course suggests that BTSP may be similar to the above-mentioned three-factor forms of plasticity, with synaptic activity generating local signals marking synapses as eligible for plasticity (ETs), and plateau potentials acting as the delayed factor that converts synaptic ETs into changes in synaptic strength. However, BTSP was shown to strengthen many synaptic inputs whose activation did not coincide with any postsynaptic spiking or even subthreshold depolarization detected at the soma (*Bittner et al., 2017*), suggesting that changes in synaptic weight might be independent of correlated pre- and postsynaptic activity, and that BTSP may be fundamentally different than all variants of Hebbian synaptic plasticity (*Gerstner et al., 2018*; *Keck et al., 2017*;

*Shouval et al., 2010*; *Mehta, 2004*; *Golding et al., 2002*). Such a non-standard plasticity rule could enable learning to be guided by delayed behavioral outcomes, rather than by short timescale associations of pre- and postsynaptic activity.

In this study, we tested the effect of dendritic plateau potentials on the spatial selectivity of CA1 neurons that already express pre-existing place fields, and therefore exhibit substantial postsynaptic depolarization and spiking prior to plasticity induction. We found that dendritic plateau potentials rapidly translocate the place field position of hippocampal place cells, both by strengthening inputs active near the plateau position and weakening inputs active within the original place field. In order to determine if the increased postsynaptic activity in place cells is causally related to the synaptic depression observed within the initial place field, we performed a series of voltage perturbation experiments, which indicated that the direction of plasticity induced by plateau potentials is independent of postsynaptic depolarization and spiking. Next, we inferred from the data a computational model of the synaptic learning rule underlying this bidirectional form of plasticity, which suggested that it is instead the current weight of each synaptic input that controls the direction of plasticity such that weak inputs potentiate and strong inputs depress. Finally, we implemented this weight-dependent learning rule in a network model to explore the capabilities of bidirectional BTSP to adapt network-level population representations to changes in the environment.

## Results

### Plateau potentials translocate existing place fields

We first examined how plasticity induced by dendritic plateau potentials changes the intracellular membrane potential ($V_m$) dynamics in neurons already exhibiting location-specific firing (i.e. place cells). Intracellular voltage recordings from CA1 pyramidal neurons were established in head-fixed mice trained to run for a water reward on a circular treadmill decorated with visual and tactile cues to distinguish spatial positions (~185 cm in length). Brief step currents (700 pA, 300 ms) were injected through the intracellular electrode for a small number (*Nakazawa et al., 2004*; *O'Keefe and Conway, 1978*; *Mehta et al., 1997*; *Lever et al., 2002*; *Dupret et al., 2010*; *Zaremba et al., 2017*; *Turi et al., 2019*; *Ziv et al., 2013*) of consecutive laps to evoke plateau potentials at a second location that was between 0 and 150 cm from the initial place field (labeled 'Induction 2' in *Figure 1A and B*; $n$ = 26 plasticity inductions in 24 neurons). In 8/24 neurons a 'natural' pre-existing place field was expressed from the start of recording, while in 16/24 the initial place field was first experimentally induced by the same procedure (labeled 'Induction 1' in *Figure 1A and B*). In 2/24 neurons the induction procedure was repeated a third time with plateaus evoked at a different location, resulting in a total of 26 plasticity inductions in cells with pre-existing place fields (see *Figure 1—figure supplement 1E* and Materials and methods).

In most cases the evoked dendritic plateaus shifted the location of the neuron's pre-existing place field toward the position of the second induction site (*Figure 1A and B*). Place field firing is known to be driven by a slow, ramping depolarization of $V_m$ from sub- to supra-threshold levels (*Bittner et al., 2015*; *Harvey et al., 2009*). Isolation of these low-pass filtered $V_m$ ramps (*Figure 1—figure supplement 2A-H*; Materials and methods) revealed that plateau potentials likewise shifted the neuron's $V_m$ ramp toward the position of the plateau, such that the new $V_m$ ramp peaked near the plateau position in most neurons (average distance = 19.5 ± 4.7 cm; $n$ = 26; *Figure 1C–E* and *Figure 1—figure supplement 2*; example cells shown in *Figure 1C* are indicated with matching colored arrows in *Figure 1D*). We also observed similar shifts in place field position to be induced by spontaneous, naturally occurring plateau potentials in a separate set of recordings ($n$ = 5; *Figure 1—figure supplement 2I-M*).

### Spatial extent of Vm plasticity

The spatial profile of plateau-induced $V_m$ changes ($\Delta V_m$) (*Figure 2A*) was obtained by subtracting the average $V_m$ ramp for trials occurring before plateau initiation (*Figure 1C*; before) from the average $V_m$ ramp for trials occurring after (*Figure 1C*; after). These data indicate that plateaus induced both positive and negative changes to $V_m$ ramp amplitude (*Figure 2A and B*). In general, the increases in $V_m$ depolarization peaked near the position of the plateau, while the negative changes peaked near the initial place field (*Figure 2A and B*, and *Figure 2—figure supplement 1A and B*). Although these changes varied considerably in magnitude across cells, the peak change in the positive direction

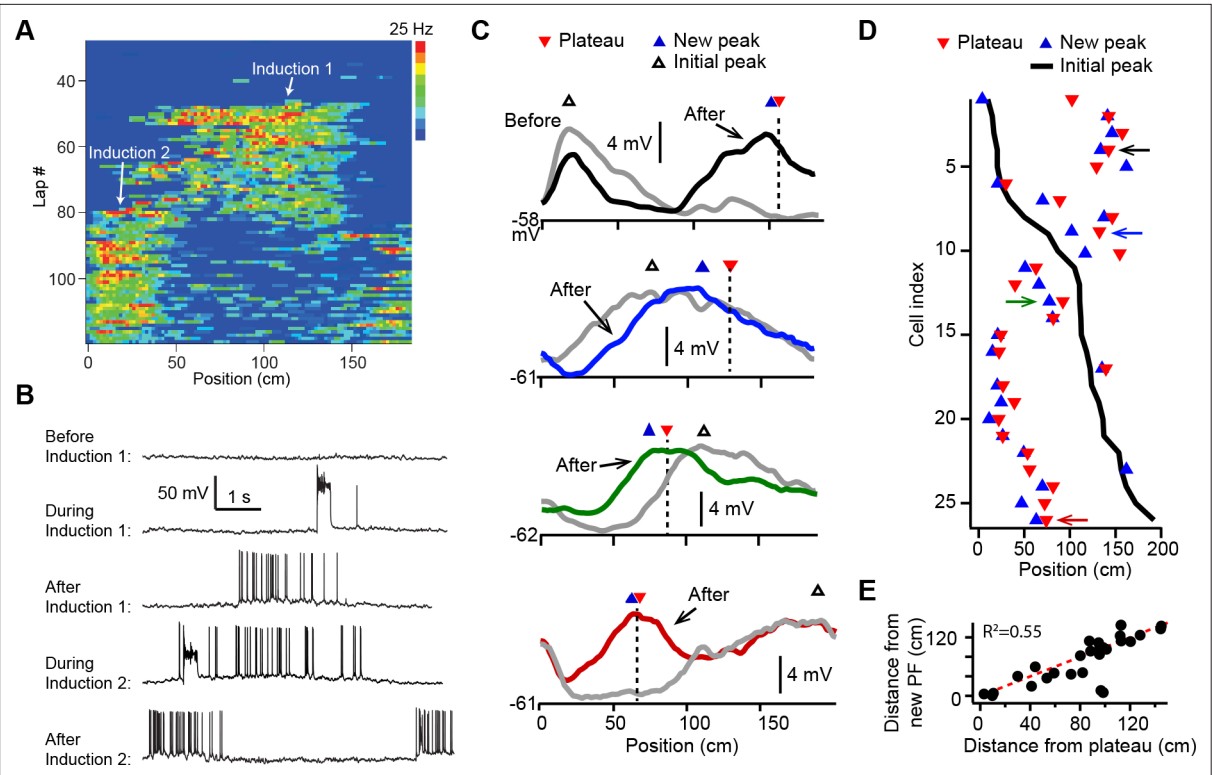

**Figure 1.** Dendritic plateau potentials translocate hippocampal place fields. (**A**) Spatial firing of a CA1 pyramidal cell recorded intracellularly from a mouse running laps on a circular treadmill. Dendritic plateau potentials evoked by intracellular current injection first induce a place field at ~120 cm (Induction 1), then induce a second place field at ~10 cm and suppress the first place field (Induction 2). (**B**) Intracellular $V_m$ traces from individual laps in (**A**). (**C**) Spatially binned $V_m$ ramp depolarizations averaged across 10 laps before (gray) and after (black, blue, green, red) the second induction (100 spatial bins). Dashed lines and red triangles mark the average locations of evoked plateaus, black open triangles mark the location of the initial $V_m$ ramp peak, and blue triangles indicate the position of the peak of the new place field. (**D**) Data from all cells were sorted by position of initial place field. Black line indicates location of initial peak, blue triangles indicate the position of the peak of the new place field, and red triangles the position of the plateau. Neurons in (**C**) are indicated by like colored arrows. (**E**) The distance between the new place field and the initial place field vs. the distance between the plateau and the initial place field (p = 0.000015; two-tailed null hypothesis test; explained variance [$R^2$] computed by Pearson's correlation). Red line is unity.

The online version of this article includes the following figure supplement(s) for figure 1:

**Figure supplement 1.** Animal run behavior and behavioral timescale synaptic plasticity (BTSP) induction procedures (related to *Figures 1 and 2*).

**Figure supplement 2.** Characterization of behavioral timescale synaptic plasticity (BTSP)-induced changes in $V_m$ (related to *Figure 1*).

was greater than the peak change in the negative direction (mean positive change± SEM vs. mean negative change± SEM: 6.73 ± 0.73 mV vs. 3.89 ± 0.32 mV, n = 26 inductions; p = 0.0001, paired two-way Student's t-test; *Figure 2—figure supplement 1A*). Aligning each $\Delta V_m$ trace to the position of the plateau (*Figure 2A and B*) demonstrates that the increases in $V_m$ depolarization observed near the plateau position decay with distance, eventually becoming hyperpolarizing decreases in $V_m$. At even greater distances from a plateau, $\Delta V_m$ decays back to zero (*Figure 2B*). To summarize the data presented thus far, dendritic plateau potentials change the location of place field firing by depolarizing $V_m$ around the plateau position and hyperpolarizing $V_m$ at positions within a pre-existing place field.

## Time dependence of Vm plasticity

Previously we showed that location-specific increases in $V_m$ depolarization induced by plateau potentials are the result of synapse-specific increases in the strength of spatially tuned excitatory inputs (*Bittner et al., 2017*). The above results suggest that, in addition to this synaptic potentiation, BTSP is also capable of inducing synaptic depression to cause location-specific decreases in $V_m$ depolarization. In analyzing the spatial extent of the $V_m$ changes induced by plateaus, we observed a strong

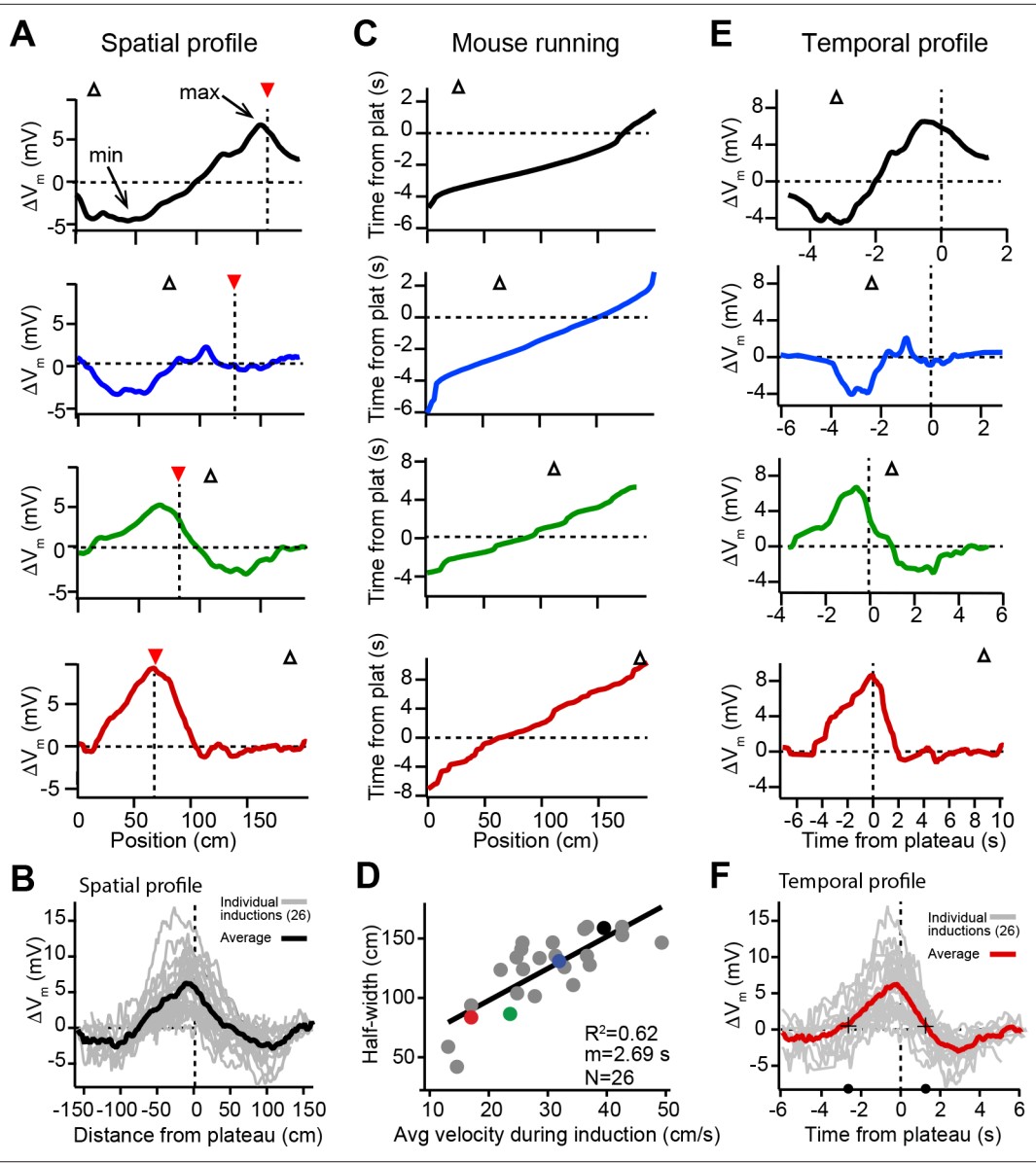

**Figure 2.** Spatial and temporal profiles of plateau-induced change in $V_m$. (**A**) Difference between spatially binned $V_m$ ramp depolarizations averaged across laps after the second induction and those averaged across laps before the second induction. Same example traces as shown in *Figure 1C*. Red triangles and dashed line indicate plateau location. Open triangles are locations of initial $V_m$ ramp peaks. Traces have been smoothed using a five point boxcar average. (**B**) All change in $V_m$ traces ($\Delta V_m$, not smoothed) from individual neurons (gray) and averaged across cells (black). (**C**) The running profile of the mice during the plateau induction trials plotted as time from plateau initiation vs. spatial location (100 bins). This indicates the temporal distance of the mouse from the plateau position at any given spatial position and is used as a time base in (**E**) and (**F**). (**D**) Spatial $V_m$ ramp half-width, calculated as distance from plateau position to the final decay of $\Delta V_m$ in a single direction, vs. the average running speed of the mouse during the induction trials calculated from traces shown in (**C**). Individual symbols for examples shown in (**A**) are correspondingly colored. Gray line is linear fit (p = 1.8e-06, two-tailed null hypothesis test; explained variance ($R^2$) computed by Pearson's correlation). (**E**) Change in $V_m$ traces ($\Delta V_m$) using the time base shown in (**C**). Traces have been smoothed using a five-point boxcar average. (**F**) All change in $V_m$ traces ($\Delta V_m$, not smoothed) from individual neurons (gray) and averaged across cells (red). Black crosses and circles indicate the 10% peak amplitude times used to calculate the asymmetry of positive changes (left/right potentiation ratio).

The online version of this article includes the following figure supplement(s) for figure 2:

**Figure supplement 1.** $V_m$ changes in space and time in place cells with pre-existing place fields (related to *Figure 2*).

linear relationship between the width of the resulting $\Delta V_m$ and the running speed of the animal during plateau induction laps (*Figure 2C and D*), which had a slope on the order of seconds. This suggested that the run trajectory of the animal (*Figure 2C*) affected the spatial extent of the plasticity (*Figure 2A and B*) by determining which positions were traversed within a fixed seconds-long temporal window for plasticity, as we previously reported (*Bittner et al., 2017*). Therefore, we next analyzed the temporal relationship between plateau potentials and location-specific potentiation and depression. To do this, we used the running trajectory of the mice during plateau induction trials (*Figure 2C*) as a time base for $\Delta V_m$ (*Figure 2E*; see also *Figure 1—figure supplement 1*, *Figure 1—figure supplement 2A-H* and Materials and methods). This analysis showed that the positive and negative changes to $V_m$ induced in place cells occurred over a timescale of multiple seconds (*Figure 2F*), with the positive changes appearing to be asymmetric with respect to the onset time of the plateaus (ratio of potentiation duration before/after plateau onset: 2.2; black circles and crossmarks in *Figure 2F* mark the time points when $\Delta V_m$ crosses zero). This asymmetry was similar to that observed for the positive $V_m$ changes induced by BTSP in silent cells (*Bittner et al., 2017*). The negative changes (i.e. the hyperpolarizations indicative of synaptic depression) occurred within a time window between ±2 and ±6 s from the plateau in many neurons that expressed pre-existing place fields (*Figure 2E and F*). Notably, this hyperpolarization was greatly reduced, or even absent, in a set of place cells where the time delay between plateau onset and the initial place field $V_m$ ramp was greater than 4–5 s (red traces in *Figures 1C, 2A and E*; see also *Figure 2—figure supplement 1C-F*), further indicating the time delimited aspect of the depression component. These data reinforce the idea that BTSP is a bidirectional form of synaptic plasticity with a seconds-long timescale that enables dendritic plateau potentials to shift the locations of hippocampal place fields by inducing both synaptic potentiation and depression.

## Plasticity drives Vm towards a target shape with an apparent inverse dependence on initial Vm

We next sought to understand why dendritic plateaus induce both $V_m$ depolarization and $V_m$ hyperpolarization in cells expressing pre-existing place fields (*Figures 1 and 2*), but induce only $V_m$ depolarization in spatially untuned silent cells (*Figure 3—figure supplement 1*; *Bittner et al., 2017*). *Figure 3A* shows that the initial temporal profile of $V_m$ in place cells with pre-existing place fields was highly variable across neurons, as plateaus were experimentally induced at different temporal intervals from the existing place field in different neurons. In contrast, the change in $V_m$ ($\Delta V_m$) induced by plateaus showed a more consistent shape in time that appeared to depend on the initial level of $V_m$ depolarization at each time point prior to plasticity (*Figure 3B*). Large positive changes occurred at time points with relatively hyperpolarized initial $V_m$, while time points with more depolarized initial $V_m$ were associated with less positive and more negative $\Delta V_m$. These changes resulted in final $V_m$ profiles that were highly similar across neurons, regardless of the initial $V_m$ (*Figure 3C*). These results indicate that BTSP induces variable changes in synaptic strength that reshape the selectivity of neurons toward a common target shape – a place field centered near the location of evoked plateau potentials that decays toward baseline over many seconds in each direction.

In *Figure 3D-F*, we examined this further by comparing data from initially hyperpolarized silent cells (black; $n = 29$ inductions, see *Figure 3—figure supplement 1* and Materials and methods) to data from place cells (dark red; $n = 26$ inductions). Place cells were on average more depolarized before plasticity than silent cells (*Figure 3D*), and more depression occurred in place cells compared to silent cells (*Figure 3E*). However, each place cell had both spatial positions where it was depolarized within its place field, and positions where it was hyperpolarized out-of-field. To determine if spatial positions that were initially depolarized were associated with larger depression, we grouped $V_m$ ramp data from all place cells, considering only spatial bins where each cell was more depolarized than a threshold of –56 mV (light red traces labeled 'PCs (within-field)' in *Figure 3D-F*). Indeed, more depression and less potentiation was induced in place cells at those spatial positions that were initially most depolarized (*Figure 3E*). However, the final $V_m$ ramps after plasticity were less sensitive to the initial state of depolarization across spatial bins of place cells (*Figure 3F*). This analysis further supported the findings that, while changes in $V_m$ induced by plateaus were highly dependent on initial $V_m$, these changes drove the resulting final $V_m$ ramp toward a common target shape (*Figure 3F*). Indeed, when all spatial bins from all place cells were analyzed, $\Delta V_m$ showed a strong inverse correlation with initial $V_m$ ($m =$

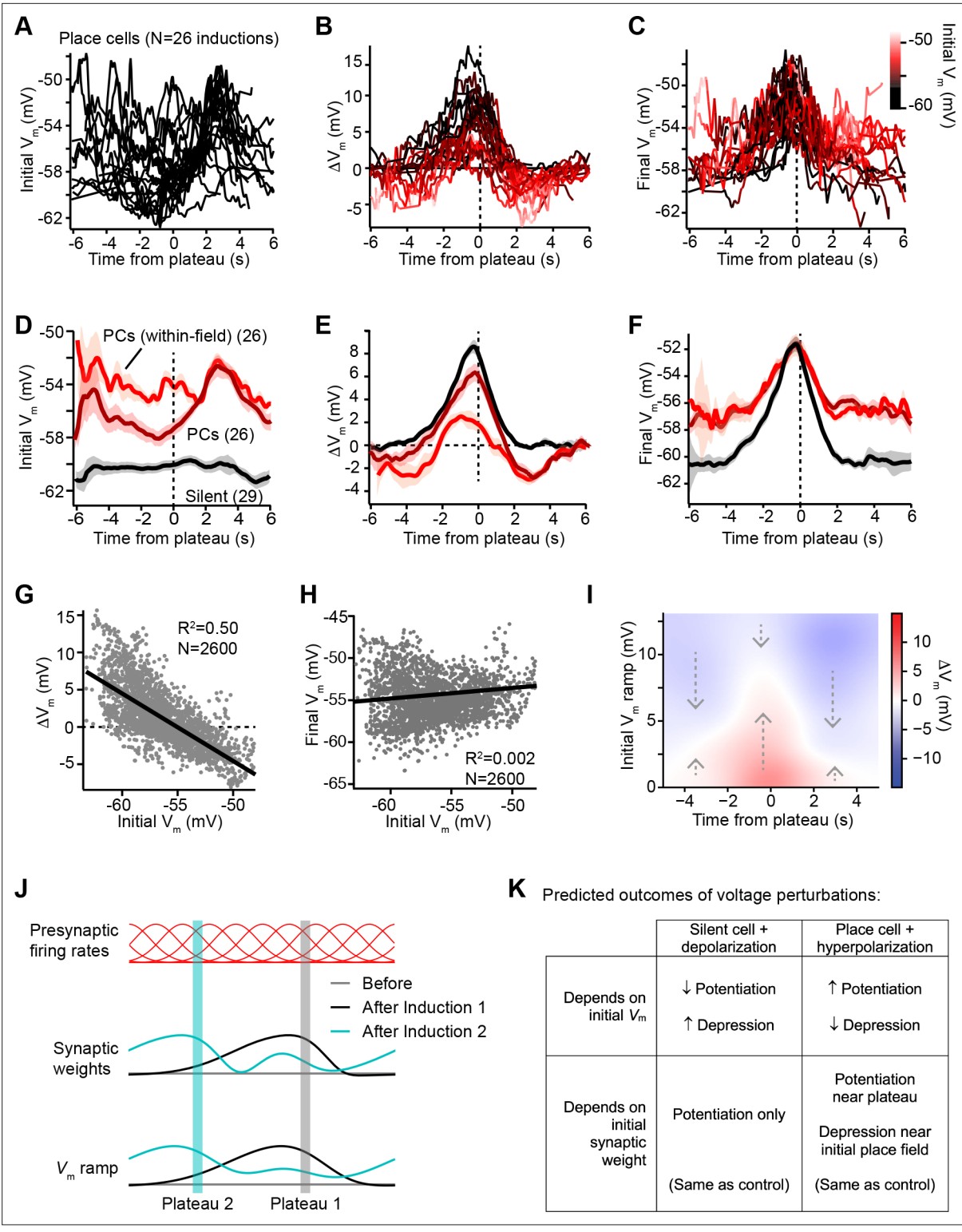

**Figure 3.** $V_m$ ramp plasticity varies with both time delay from plateau onset and initial $V_m$ depolarization. (**A**) Temporal profile of initial $V_m$ before plasticity for inductions in neurons with pre-existing place fields (26 inductions from 24 place cells), aligned to the onset time of evoked plateau potentials. (**B**) Temporal profile of changes in $V_m$ ($\Delta V_m$) induced by plasticity in all place cells. Each $\Delta V_m$ trace is color-coded by initial $V_m$. See inset color scale in (**C**). (**C**) Temporal profile of final $V_m$ after plasticity in all place cells. Each $V_m$ trace is color-coded by initial $V_m$ (color scale inset). (**D**) The temporal profiles of initial $V_m$ before plasticity are averaged across cells and three conditions are compared: silent cells without pre-existing place fields (black), place cells (dark red), and a subset of data from each place cell at time points when each cell was more depolarized than −56 mV within its place field

*Figure 3 continued*

(light red). Shading indicates SEM across cells. (**E**) The temporal profiles of changes in $V_m$ ($\Delta V_m$) induced by plasticity are averaged across cells and the three conditions from (**D**) are compared. Shading indicates SEM across cells. (**F**) The temporal profiles of final $V_m$ after plasticity are averaged across cells and the three conditions from (**D**) are compared. Shading indicates SEM across cells. (**G**) Change in $V_m$ ramp ($\Delta V_m$) plotted against initial $V_m$ for all inductions in neurons with pre-existing place fields. Black line is linear fit and correlation coefficient shown (p < 0.00001, two-tailed null hypothesis test; explained variance ($R^2$) computed by Pearson's correlation). (**H**) Final $V_m$ ramp after plasticity plotted against initial $V_m$ before plasticity for all inductions in neurons with pre-existing place fields. Black line is linear fit and correlation coefficient shown (p < 0.014, two-tailed null hypothesis test; explained variance ($R^2$) computed by Pearson's correlation). (**I**) Heatmap of changes in $V_m$ ramp ($\Delta V_m$) as a function of both time and initial $V_m$ (see Materials and methods). Arrows indicate that the variable direction of plasticity serves to drive $V_m$ toward the target equilibrium region (white). (**J**) Diagram depicts presynaptic spatial firing rates of a population of CA3 inputs to a postsynaptic CA1 neuron (top), the synaptic weights of those inputs before and after plasticity (middle), and the resulting postsynaptic $V_m$ ramp, which reflects a weighted summation of the inputs. Traces are shown before (gray) and after (black) plasticity induction in a silent cell (Induction 1), and after a subsequent induction of plasticity (Induction 2, cyan) that translocates the position of the cell's place field. (**K**) Table compares predicted outcomes of voltage perturbation experiments (depolarizing a silent cell, or hyperpolarizing a place cell), considering two possible forms of behavioral timescale synaptic plasticity (BTSP) (depends on initial $V_m$, or depends on initial synaptic weights).

The online version of this article includes the following figure supplement(s) for figure 3:

**Figure supplement 1.** $V_m$ changes in space and time in silent cells without pre-existing place fields (related to *Figure 3*).

–0.91; *Figure 3G*). In contrast, final $V_m$ showed a very weak positive correlation with initial $V_m$ (m = 0.04; *Figure 3H*), which reflects that some spatial bins show no change in $V_m$ during plasticity, either because they were traversed outside the temporal window for plasticity or because the $V_m$ at those positions had already reached a final $V_m$ target value.

That BTSP induces variable changes in $V_m$ that reshape the $V_m$ ramp toward a particular target shape is further evident from a heatmap depicting the relationships of $\Delta V_m$ to both initial $V_m$ ramp depolarization and time from plateau onset (*Figure 3I*, positive $\Delta V_m$ in red, and negative $\Delta V_m$ in blue; see Materials and methods). The white regions of this plot trace out a temporal profile of $V_m$ that corresponds to the final target place field shapes shown in *Figure 3C and F*. All initial deviations from this equilibrium $V_m$ profile resulted in either positive or negative changes to approach this target place field shape (see dashed arrows). It should also be noted that the depression of $V_m$ in place cells appeared to be weaker than the potentiation, leaving some residual depolarization at positions distant from the peak (*Figure 3F*). The functional significance of this is unclear, but may suggest that BTSP induces synaptic depression at a slower rate than potentiation (*Cone and Shouval, 2021*). To summarize, BTSP induces precise changes in synaptic strength that modify pre-existing place fields with any initial shape such that they approach a target shape that peaks near the location where dendritic plateaus were evoked.

## Dependence on initial *Vm* vs. initial synaptic weights

Altogether these data revealed that, in general, the magnitude and direction of $\Delta V_m$ depended on the time from the plateau potential, and correlated inversely with the initial $V_m$ ramp amplitude prior to plasticity induction. Does this anti-correlation reflect a causal relationship between postsynaptic depolarization and changes in synaptic weight induced by BTSP? This possibility would require that small depolarizations induce synaptic potentiation and large depolarizations induce synaptic depression, which is actually opposite to what has been observed in CA1 pyramidal cells with a variety of other plasticity protocols (*Shouval et al., 2010*; *Yang et al., 1999*; *Graupner and Brunel, 2012*; *Clopath et al., 2010*; *Clopath and Gerstner, 2010*; *Jedlicka et al., 2015*). Furthermore, the increased $V_m$ depolarization within a cell's place field also reflects the activation of strongly weighted synaptic inputs, which have been potentiated by prior plasticity (*Bittner et al., 2015*; *Bittner et al., 2017*; *Figure 3J*). Thus, a causal dependency on either $V_m$ or synaptic weight could explain the data so far.

To discriminate between these two possibilities, we next devised a set of voltage perturbation experiments. We reasoned that, if increased depolarization and spiking within a cell's place field causes synaptic depression, then artificially increasing $V_m$ and inducing spiking in otherwise silent cells would cause plateau potentials to induce negative $\Delta V_m$. Likewise, artificially decreasing $V_m$ and preventing spiking in place cells would prevent plateau potentials from inducing negative $\Delta V_m$ (*Figure 3K*). On the contrary, if the direction of plasticity depended instead on the initial strengths of synapses prior to plasticity, these voltage manipulations would have no effect on the balance between positive and negative $\Delta V_m$ (*Figure 3K*). It is important to note that these somatic voltage

manipulations are not expected to strictly control or even completely overwhelm $V_m$ at the synaptic sites relevant to plasticity induction (*Magee and Johnston, 1997*; *Koester and Sakmann, 1998*; *Froemke et al., 2005*) due to attenuation of current and voltage along the dendritic cable (*Magee, 1998*; *Golding et al., 2005*), and compartmentalization of synaptic voltage in dendritic spines (*Harnett et al., 2012*). However, by either increasing or decreasing the generation of somatic action potentials, this manipulation will unequivocally alter the number of action potentials that back-propagate into dendrites, which will in turn influence the activation of voltage-gated channels in dendrites and spines (e.g. $Na^+$ channels, $Ca^{2+}$ channels, and NMDA-Rs) (*Magee and Johnston, 1997*; *Takahashi and Magee, 2009*). Expected changes to the mean $V_m$ in active dendritic spines were supported by simulations of a biophysically and morphologically detailed CA1 place cell model expressing voltage-gated ion channels and receiving rhythmic excitation and inhibition to mimic the in vivo recording conditions (*Figure 4—figure supplement 1*; *Grienberger et al., 2017*). Moreover, manipulation of somatic $V_m$ and spike timing is widely used to successfully influence plasticity induction in vitro and in vivo (*Malinow and Miller, 1986*; *Jacob et al., 2007*; *Schulz et al., 2010*).

According to the above scheme, we first recorded from spatially untuned silent cells, and injected current (~100 pA) through the intracellular pipette to depolarize the neurons' $V_m$ by ~10 mV and to increase spiking during plasticity induction trials (*Figure 4A*; baseline trials mean AP rate: 0.26 ± 0.25 Hz; first induction trial mean AP rate: 4.8 ± 1.4 Hz, $n$ = 8; blue trace in *Figure 4B*). In all neurons tested, we observed plateau potentials to induce large positive $\Delta V_m$ at spatial positions surrounding the plateau location, and no negative $\Delta V_m$ at any spatial positions (*Figure 4A*; blue trace in *Figure 4C*). This result is inconsistent with a causal dependence on initial $V_m$ (*Figure 3K*), which predicted a $\Delta V_m$ profile similar to that of control place cells at their most depolarized positions within their pre-existing place fields (red traces, 'control PCs (within-field)' in *Figure 4B and C* repeated from *Figure 3D and E* for comparison).

Next, we performed the inverse manipulation by recording from place cells and injecting current (~–150 pA) to hyperpolarize the neurons' $V_m$ by ~–15 mV and prevent spiking at spatial locations surrounding their pre-existing place fields while plasticity was induced at a second location (*Figure 4D*; baseline trials in-field mean AP rate 10.66 ± 0.93 Hz; first induction trial in-field mean AP rate 0.06 ± 0.06 Hz, $n$ = 5; green trace in *Figure 4E*). This manipulation did not prevent negative $\Delta V_m$ at positions within the original place field (*Figure 4D*; green trace in *Figure 4F*), again incompatible with synaptic depression requiring elevated postsynaptic depolarization and spiking (*Figure 3K*). In fact, full amplitude synaptic depression was observed at locations within the original place field despite the somatic $V_m$ being more hyperpolarized than either the silent cell (black traces in *Figure 4E and F*) or control place cell groups (red traces, 'control PCs' in *Figure 4E and F*).

These data clearly show that the direction of plasticity induced by dendritic plateau potentials is not determined by the activation state of the postsynaptic neuron. Instead, the results of these voltage perturbation experiments support the alternative hypothesis that it is the initial strength of each synapse that controls whether an input will be potentiated or depressed by BTSP (*Figure 3K*). However, the magnitude of potentiation and depression was slightly affected by the voltage perturbations (e.g. potentiation was slightly but significantly increased in silent cells during artificial depolarization compared to control, *Figure 4C*). This is consistent with the previously reported finding that BTSP induction requires activation of voltage-dependent ion channels, including NMDA-type glutamate receptors (NMDA-Rs) and voltage-gated calcium channels (*Bittner et al., 2017*), which would have predicted BTSP to depend on postsynaptic depolarization. To examine this further, we performed an additional set of experiments in which silent cells were strongly hyperpolarized by somatic current injection (~–50 mV for ~3 s just before plateau initiation) during plasticity induction (*Figure 4—figure supplement 2*). This manipulation decreased synaptic potentiation (*Figure 4—figure supplement 2*), consistent with a requirement for activation of voltage-dependent NMDA-Rs. That such a large, non-physiological level of global $V_m$ hyperpolarization was required to alter BTSP reinforces the finding that, operationally, the dependence is not on voltage signals associated with neuronal activation state (sustained somatodendritic $V_m$ and action potentials), but rather on those associated with synaptic input (transient local spine depolarization) (*Beaulieu-Laroche and Harnett, 2018*). Finally, these experiments do not support a role for synaptic depolarization in determining the *direction* of changes in synaptic strengths.

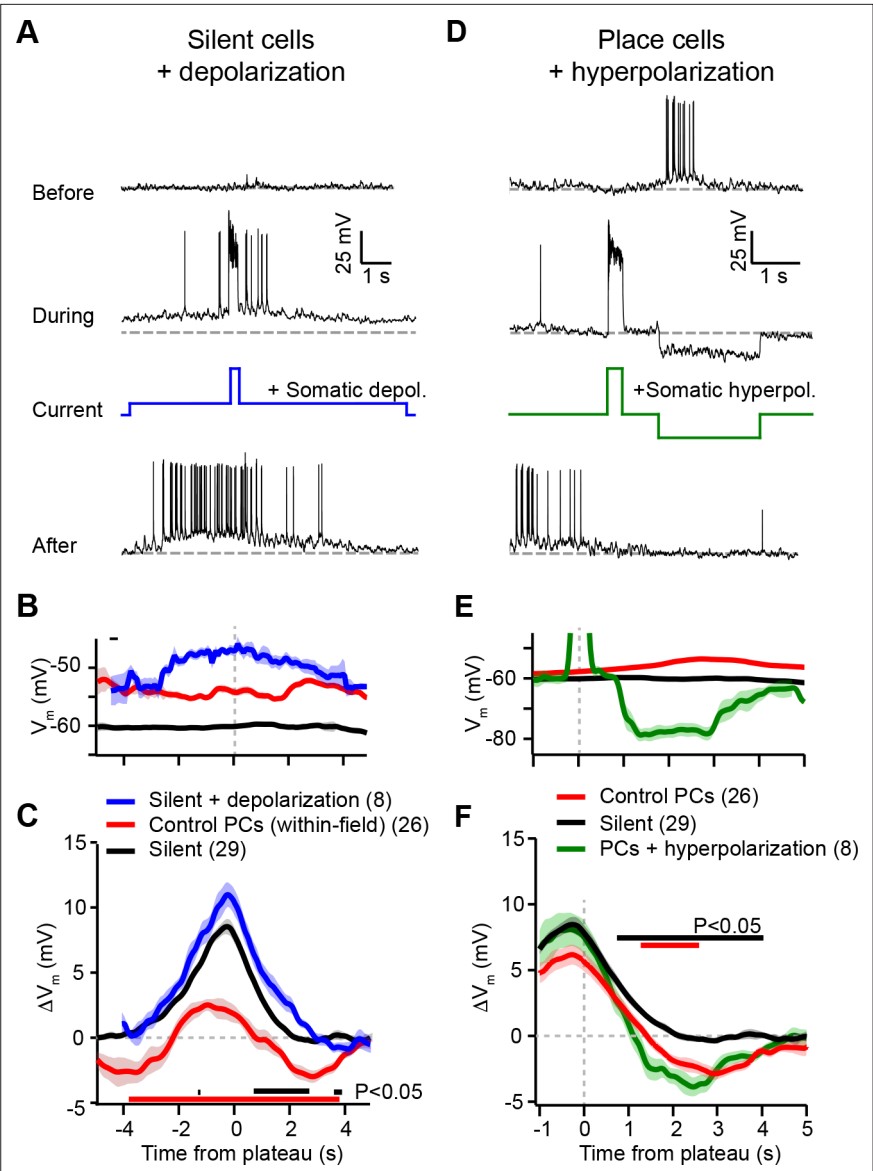

**Figure 4.** Experimental perturbation of postsynaptic activation does not change the direction of plasticity induced by behavioral timescale synaptic plasticity (BTSP). (**A**) Intracellular $V_m$ traces from individual laps in which plasticity was induced by experimentally evoked plateau potentials in an otherwise silent CA1 cell (top trace). During plasticity induction laps (middle trace), the neuron was experimentally depolarized by ~10 mV. Experimentally evoked plateau potentials induced a place field (bottom). (**B**) Initial $V_m$ before plasticity averaged across cells. Shading indicates SEM. Three conditions are compared: manipulated silent cells (silent + depolarization; blue), data from place cells at time points within their initial place fields (control PCs [within-field]; red), and control cells without pre-existing place fields (silent; black). (**C**) Changes in $V_m$ ramp ($\Delta V_m$) for the same groups as in (**B**). Colored bars indicate statistical significance in specific time bins (p < 0.05; Student's two-tailed t-test). Black compares manipulated silent cells to control silent cells, and red compares manipulated silent cells to control place cells (within-field). See Materials and methods for number of inductions in each time bin. (**D**) Intracellular $V_m$ traces from individual laps in which plasticity was induced by experimentally evoked plateau potentials in a place cell with a pre-existing place field (top). During plasticity induction laps, the neuron was experimentally hyperpolarized by ~25 mV at spatial positions surrounding the initial place field (middle). Experimentally evoked plateau potentials translocated the place field (bottom). (**E**) Initial $V_m$ before plasticity averaged across cells. Shading indicates SEM. Three conditions are compared: control place cells with pre-existing place fields (control PCs; red), control silent cells without pre-existing place fields (silent; black), and manipulated place cells with pre-existing place fields (PCs + hyperpolarization; green). (**F**) Changes in $V_m$ ramp ($\Delta V_m$) for the same groups as in (**E**). Colored bars indicate statistical significance in specific time bins (p < 0.05; Student's two-tailed t-test). Black compares manipulated place

*Figure 4 continued on next page*

*Figure 4 continued*

cells to control silent cells, and red compares manipulated place cells to control place cells. See Materials and methods for number of inductions in each time bin.

The online version of this article includes the following figure supplement(s) for figure 4:

**Figure supplement 1.** Biophysically detailed simulations of depolarizing and hyperpolarizing somatic $V_m$ perturbation experiments (related to *Figure 4*).

**Figure supplement 2.** Hyperpolarization of silent cells during plasticity induction reduces synaptic potentiation (related to *Figure 4*).

## Weight-dependent model of bidirectional BTSP

The above voltage perturbation experiments suggested that the form of synaptic plasticity underlying BTSP does not depend on the activation state of the postsynaptic neuron (*Figure 4*). This contrasts with Hebbian plasticity rules that typically depend on either the firing rate or depolarization of the postsynaptic cell to determine the amplitude and direction of changes in synaptic weight. Another difference is that BTSP appears to be inherently stable, converting synaptic potentiation into depression when input strengths exceed a particular range, whereas most models of Hebbian learning require additional homeostatic mechanisms to counteract synaptic potentiation in highly active neurons (*Oja, 1982*; *Bienenstock et al., 1982*; *Abbott and Nelson, 2000*; *Zenke et al., 2013*; *Turrigiano and Nelson, 2004*). To better understand the synaptic learning rule underlying BTSP and its functional consequences, we next sought a mathematical description of BTSP to account for the following features of the in vivo recording data:

1. BTSP induces bidirectional changes in synaptic weight at inputs activated up to ~6 s before or after a dendritic plateau potential.
2. The direction and magnitude of changes in synaptic weight depend on the initial state of each synapse such that weak inputs potentiate, and strong inputs depress.
3. BTSP modifies synaptic weights such that the temporal profile of Vm in place cells approaches a stable target shape that peaks close in time to the plateau location and decays with distance.

As mentioned previously, 'three-factor' plasticity models propose a mechanism for the strengths of activated synapses to be modified after a time delay – a biochemical intermediate signal downstream of synaptic activation marks each recently activated synapse as 'eligible' to undergo a plastic change in synaptic weight. This 'ET' decays over a longer timescale than synaptic activation, and while it does not induce plasticity by itself, it enables plasticity to be induced upon the arrival of an additional modulatory biochemical signal. While 'three-factor' models consider synaptic ETs to be generated by a coincidence of presynaptic spikes (factor 1) and postsynaptic spikes or sustained depolarization (factor 2), the results of the above voltage perturbation experiments suggest that if BTSP involves the generation of synaptic ETs, these signals depend only on a single factor – local synaptic activation. In the context of BTSP, the modulatory or 'instructive signal' (IS) could be instantiated by a dendritic plateau potential. To model this, we assumed that the large magnitude dendritic depolarization associated with a plateau potential (~60 mV) effectively propagates to all synapses (*Xu et al., 2012*), activating an IS at each synapse and allowing a spatially and temporally local interaction between ET and IS to drive plasticity independently at each individual synapse (*Figure 5A*). To account for plasticity that occurs at inputs activated up to multiple seconds *after* a plateau, this IS would have to decay slowly enough to overlap in time with ETs generated after the end of the plateau (*Figure 5A*).

Accordingly, we modeled changes in synaptic weights as a function of the time-varying amplitudes of these two biochemical intermediate signals, ET and IS. For simplicity, we first considered how BTSP would change the weight $W$ of a single synapse activated by a single presynaptic spike with precise timing relative to the onset of a plateau potential (*Figure 5A*). We modeled the synaptic ET as a signal that increases upon synaptic activation at time $t^s$ and decays exponentially with time course $\tau_{ET}$ (see *Figure 5A* and Materials and methods). The IS was modeled as a signal that increases during a plateau potential with onset at time $t^p$ and duration $d$ and decays exponentially with time course $\tau_{IS}$ (see *Figure 5A* and Materials and methods).

Next, we modeled bidirectional changes in synaptic weight $\frac{dW}{dt}$ as a function of the temporal overlap or product of these two signals, $ET * IS$. To account for the observation that BTSP favors synaptic potentiation at weak synapses and synaptic depression at strong synapses, we expressed $\frac{dW}{dt}$

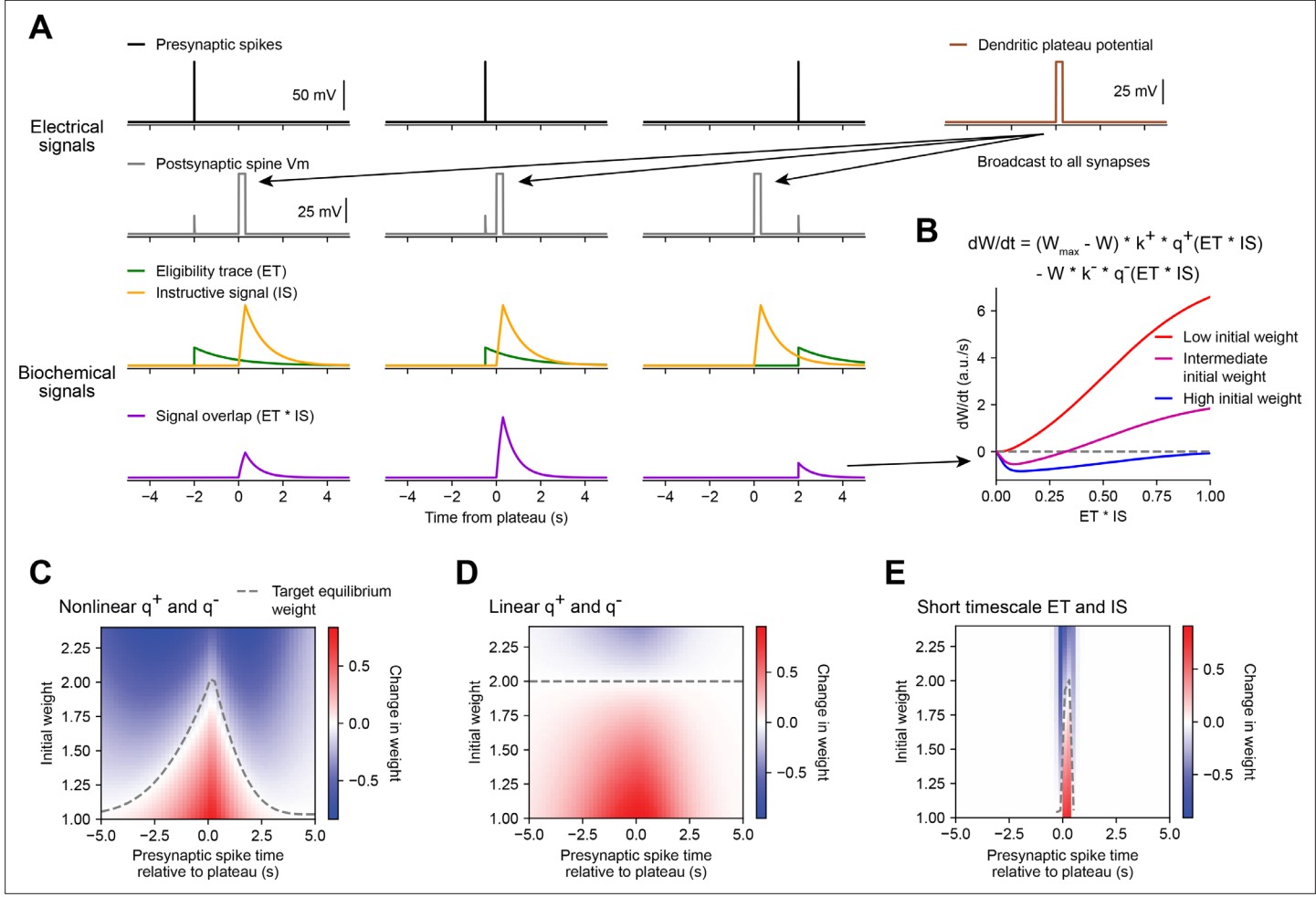

**Figure 5.** Weight-dependent model of behavioral timescale synaptic plasticity (BTSP) captures essential features of plateau-induced plasticity. (**A – B**) Traces schematize a model of bidirectional BTSP that depends on (1) presynaptic spike timing, (2) plateau potential timing and duration, and (3) the current synaptic weight of an input before an evoked plateau. (**A**) Presynaptic spikes (first row, black) result in local $V_m$ depolarization of a postsynaptic spine (second row, gray), which generates a long duration plasticity 'eligibility trace' (ET) (third row, green) that marks the synapse as eligible for later synaptic potentiation or depression. The large $V_m$ depolarization associated with the dendritic plateau potential (first row, brown) is assumed to effectively propagate to all synaptic sites (second row, gray), which generates a separate long duration 'instructive signal' (IS) (third row, yellow) that is also required for plasticity. Both potentiation and depression are saturable processes that depend on the temporal overlap (product) of ET and IS (fourth row, purple). (**B**) Equation defines the rate of change in synaptic weight $\frac{dW}{dt}$ in terms of a potentiation process $q^+$ that decreases with increasing initial weight $W$, and a depression process $q^-$ that increases with increasing initial weight $W$. Plot shows the relationship between $\frac{dW}{dt}$ and the signal overlap $ET * IS$ under conditions of low (red), intermediate (purple), or high (blue) initial weight. (**C – E**) Heatmaps of changes in synaptic weight in terms of time delay between presynaptic spike and postsynaptic plateau, and initial synaptic weight for three variants of the weight-dependent model of BTSP. Dashed traces mark the equilibrium initial synaptic weight at each time delay where potentiation and depression are balanced and additional pairings of presynaptic spikes and postsynaptic plateaus result in zero further change in synaptic weight. (**C**) Model in which potentiation ($q^+$) and depression ($q^-$) processes are nonlinear (sigmoidal) functions of signal overlap ($ET * IS$). (**D**) Model in which are potentiation ($q^+$) and depression ($q^-$) processes are linear functions of signal overlap ($ET * IS$). (**E**) Model in which the durations of the $ET$ and instructive signal ($IS$) are constrained to a short (100 ms) timescale, similar to intracellular calcium.

in terms of two separate plasticity processes $q^+$ and $q^-$ with opposite dependencies on the current synaptic weight $W$:

$$\frac{dW}{dt} = \left(W_{max} - W\right) * k^+ * q^+ \left(ET * IS\right) - W * k^- * q^- \left(ET * IS\right) \tag{1}$$

where $W$ is saturable up to a maximum weight of $W_{max}$, and $k^+$ and $k^-$ are learning rate constants that control the magnitudes of synaptic potentiation and depression per plateau potential. This formula can be obtained from a two-state model of finite synaptic resources (see Materials and methods).

When the current synaptic weight $W$ is near $W_{max}$, the potentiation rate becomes zero, and when $W$ is near zero, the depression rate becomes zero. To calculate the net change in synaptic weight $W$ after plasticity induction, $\frac{dW}{dt}$ was integrated in time for the duration of plasticity induction laps.

Experimental evidence suggests that synaptic potentiation and depression processes involve biochemical interactions between enzymes (e.g. phosphokinases-like CaMKII and phosphatases-like calcineurin) and synaptic protein substrates (e.g. AMPA-type glutamate receptors) (*Herring and Nicoll, 2016*; *Mansuy, 2003*). Such concentration-limited reactions are typically saturable and nonlinear (*Graupner and Brunel, 2012*). Accordingly, we defined the plasticity processes $q^+$ and $q^-$ as saturable (sigmoidal) functions of the signal overlap $ET * IS$ (see Materials and methods). If the depression process $q^-$ has a lower threshold for activation than the potentiation process $q^+$ (*Graupner and Brunel, 2007*; *Inglebert et al., 2020*), the resulting change in synaptic weight $\frac{dW}{dt}$ is positive and increases monotonically when initial weights are low, but is negative and non-monotonic when initial weights are high (*Figure 5B*). At intermediate weights, $\frac{dW}{dt}$ transitions from negative (depression) to positive (potentiation) for values of signal overlap $ET * IS$ that are beyond a threshold (*Figure 5B*). Thus, the largest negative changes in synaptic weight occur when inputs are initially large in weight and signal overlap $ET * IS$ is intermediate in amplitude. This is consistent with the in vivo data, which showed that negative changes in place field ramp $V_m$ were largest at intermediate delays from a plateau (*Figure 3B and I*).

We tested this weight-dependent model of bidirectional BTSP by varying both the timing of a single presynaptic spike relative to a plateau (*Figure 5A*) and the initial weight of the activated synapse (*Figure 5B*). Model parameters were calibrated (see Materials and methods) such that synapses with an initial weight less than a baseline weight of 1 undergo only potentiation, while synapses with higher weight undergo either potentiation or depression, depending on the timing of their activation relative to the plateau (*Figure 5C*). This produced a profile of changes in synaptic weight similar to the profile of changes in intracellular $V_m$ measured in vivo (*Figure 3I*). This model also recapitulated the finding that the positive and negative changes in weight induced by BTSP appear to drive synaptic inputs toward a stable target weight, after which additional plateaus do not induce any further changes in strength (indicated in white, compare *Figures 3I and 5C*).

We next exploited the mathematical formulation of the model to analyze these equilibrium conditions in more detail. We defined $W_{eq}$ as the stable equilibrium value of $W$ where potentiation and depression processes are exactly balanced, and the change in weight $W$ is zero over the course of a trial from times $t_0$ to $t_1$:

$$\Delta W = 0 = \left( W_{max} - W_{eq} \right) * k^+ * \int_{t_0}^{t_1} q^+ (ET * IS)\, dt - W_{eq} * k^- * \int_{t_0}^{t_1} q^- \left( ET * IS \right) dt \tag{2}$$

If we abbreviate the integrated potentiation and depression terms as:

$$\Delta Q^+ = \int_{t_0}^{t_1} q^+ (ET * IS) dt \tag{3}$$

$$\Delta Q^- = \int_{t_0}^{t_1} q^- (ET * IS) dt \tag{4}$$

then $W_{eq}$ can be expressed as:

$$W_{eq} = W_{max} * \frac{K^+ * \Delta Q^+}{K^+ * \Delta Q^+ + k^- * \Delta Q^-} \tag{5}$$

Note that the quantities $\Delta Q^+$ and $\Delta Q^-$, and therefore the value of $W_{eq}$, will vary with the activation time of the input ($t^s$), and the onset time ($t^p$) and duration ($d$) of a plateau. For a plateau with fixed onset time and duration, this produces a distribution of target equilibrium weights that varies only with the timing of synaptic activation relative to plateau onset (dashed line in *Figure 5C*), and matches the asymmetric shape of place fields induced by BTSP. In contrast, an alternative version of the model in which the potentiation and depression processes were defined to be linear instead of sigmoidal, predicted a single value for $W_{eq}$ regardless of the timing of synaptic activation (*Figure 5D*), thus failing to account for the data. Finally, we verified that the model requires long timescales for $ET$ and $IS$ by testing the model with shorter values (100 ms) for the decay time constants $\tau_{ET}$ and $\tau_{IS}$ (*Figure 5E*).

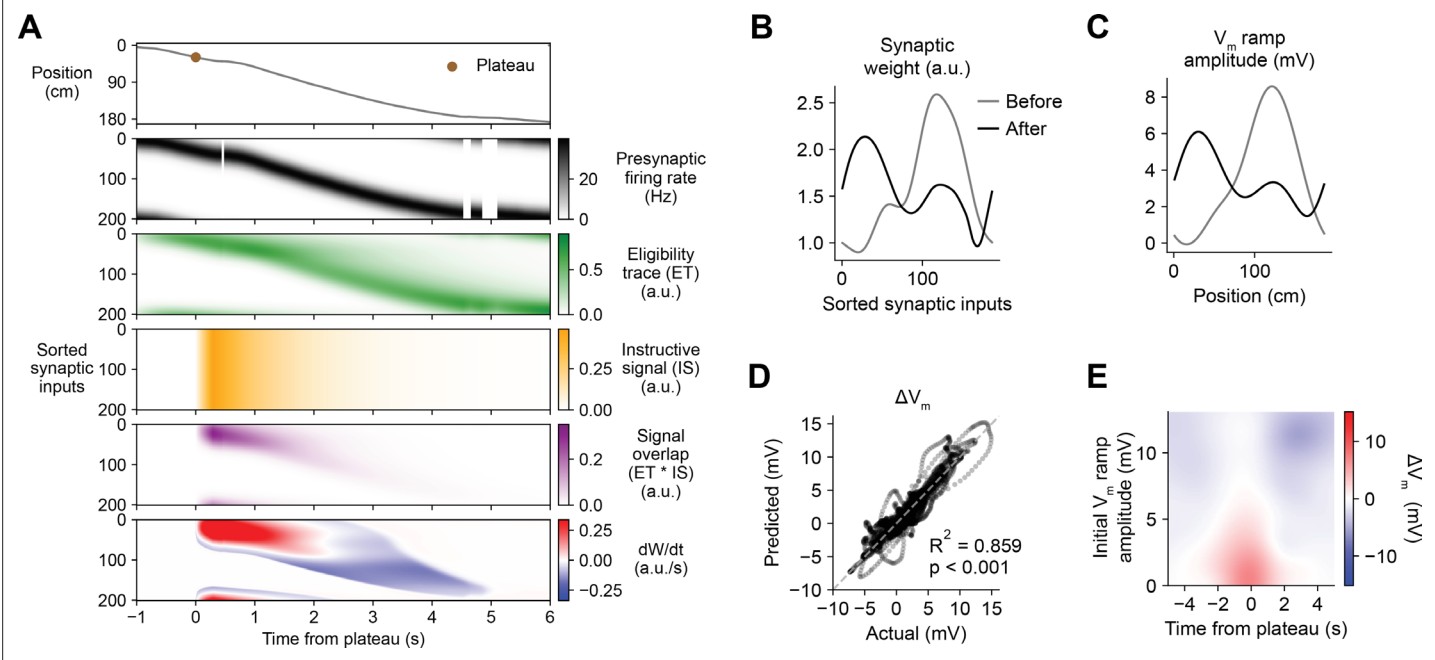

**Figure 6.** Weight-dependent model of behavioral timescale synaptic plasticity (BTSP) accounts for experimentally measured bidirectional changes in $V_m$. (**A**) The weight-dependent model of BTSP shown in **Figure 5** was used to reproduce plateau-induced changes in $V_m$ in an experimentally recorded CA1 neuron given (1) the measured run trajectory of the animal during plateau induction trials (example lap shown in first row, animal position in gray), (2) the measured timing and duration of evoked plateau potentials (first row, example plateau onset marked in brown), and (3) the measured initial $V_m$ before plasticity (shown in (**C**), gray). A population of 200 presynaptic CA3 place cells provided input to the model CA1 neuron. The firing rates of the presynaptic inputs were assumed to vary with spatial position and run velocity (second row, all presynaptic inputs are shown sorted by place field peak location, black). Synaptic activity at each input generated a distinct local eligibility trace (ET) (third row, green). The dendritic plateau potential generated a global instructive signal broadcast to all synapses (fourth row, yellow). The overlap between ET and IS varied at each input depending on the timing of presynaptic activity (fifth row, purple). The weight-dependent model predicted increases in synaptic weight (positive rate of change, red) at some synapses with low initial weight, and decreases in synaptic weight (negative rate of change, blue) at other synapses with high initial weight. (**B**) Synaptic weights of the 200 synaptic inputs shown in (**A**) before (gray) and after (black) plateau-induced plasticity. (**C**) Spatially binned $V_m$ ramp before (gray) and after (black) plasticity was computed as a weighted sum of the input activity. (**D**) Changes in $V_m$ ramp amplitude ($\Delta V_m$) at each spatial bin predicted by the weight-dependent model are compared to the experimental data ($n = 26$ inductions from 24 neurons with pre-existing place fields). Explained variance ($R^2$) and statistical significance ($p < 0.05$) reflect Pearson's correlation and two-tailed null hypothesis tests. (**E**) Heatmap of changes in $V_m$ ramp ($\Delta V_m$) predicted by the model as a function of both time and initial $V_m$. Compare to experimental data in **Figure 3I**.

The online version of this article includes the following figure supplement(s) for figure 6:

**Figure supplement 1.** Sensitivity of induced place field $V_m$ ramps to repeated plateau potentials and run velocity in the weight-dependent model of behavioral timescale synaptic plasticity (BTSP) (related to **Figures 2 and 6**).

**Figure supplement 2.** Comparison of alternative models of behavioral timescale synaptic plasticity (BTSP) (related to **Figures 5 and 6**).

**Figure supplement 3.** Bidirectional behavioral timescale synaptic plasticity (BTSP) schematic (related to **Figures 3, 5 and 6**).

This was unable to explain changes in synaptic weight at inputs activated at seconds-long time delays to a plateau.

Having demonstrated that this weight-dependent model of plasticity at single synapses captures the essential features of BTSP, we next tested if the model can account quantitatively for the in vivo place field translocation data (**Figures 1–3**). For this purpose, we assumed that the $V_m$ ramp depolarization measured in a CA1 pyramidal cell during locomotion on the circular treadmill reflects a weighted sum of presynaptic inputs that are themselves place cells with firing rates that vary with spatial position (see **Figure 3J** and Materials and methods). As a population, the place fields of these inputs uniformly tiled the track, and the firing rate of an individual input depended on the recorded run trajectory of the animal (**Figure 6A**, first and second rows). In this case, presynaptic activity patterns were modeled as continuous firing rates rather than discrete spike times. For each cell in the experimental dataset ($n = 26$ inductions from 24 neurons, **Figures 1–3**), the initial weight $W_i$ of each presynaptic input was inferred from the recorded initial $V_m$, and the changes in weight $W_i$ during

plasticity induction laps containing evoked plateau potentials were computed as above (*Equation 1*; see Materials and methods). The relevant signals modeled for an example lap from a representative cell from the dataset are shown in *Figure 6A*. Note that, at inputs activated before the onset time of the plateau, changes in synaptic weight (bottom row) do not begin until after plateau onset when the instructive signal *IS* and the signal overlap $ET * IS$ are nonzero. The parameters of the model were optimized to predict the final synaptic weights (*Figure 6B*) and reproduce the final $V_m$ ramp (*Figure 6C*) after multiple plasticity induction laps (*Figure 6—figure supplement 1*, Materials and methods). Across all cells, these predictions quantitatively matched the corresponding experimental data (*Figure 6D*). Finally, the sensitivity of changes in $V_m$ to initial $V_m$ and time to plateau predicted by the model recapitulated that measured from the in vivo intracellular recordings (*Figure 6E* and *Figure 6—figure supplement 2*).

The above modeling results help to clarify the differences between BTSP and previously characterized forms of associative synaptic plasticity based on input-output correlations over short timescales (*Gerstner et al., 2018*; *He et al., 2015*; *Brzosko et al., 2015*; *Brzosko et al., 2017*). First, the model supports the hypothesis that a dependence on initial synaptic weight is the actual source of the observed inverse relationship between initial $V_m$ and plasticity-induced changes in $V_m$ (*Figure 3*). Second, the scaling of both potentiation and depression by synaptic weight produces a balanced form of plasticity that rapidly stabilizes during repeated inductions (*Figure 1*, and *Figure 6—figure supplement 1A and B*; *Shouval et al., 2010*; *Jedlicka et al., 2015*; *Bienenstock et al., 1982*; *Abraham, 2008*; *Cooper and Bear, 2012*). Third, the time course of BTSP is determined by temporal overlap between slow eligibility signals associated with synaptic activity and slow IS associated with plateau potentials. This selects a subpopulation of synaptic inputs activated with appropriate timing to undergo a change in synaptic strength (*Figure 6—figure supplement 3*). Finally, IS are internal signals activated by dendritic plateau potentials, rather than by spiking output, arguing that BTSP is not simply a variant of Hebbian plasticity that depends on input-output correlations over a longer timescale.

## Functional capabilities of BTSP

The above observations imply that BTSP could enable spatial representations to be shaped non-autonomously by delayed behavioral outcomes, if dendritic inputs carrying information about those outcomes are able to evoke plateau potentials (*Muller et al., 2019*). To evaluate the feasibility and implications of this theory, we next considered the conditions that are required for dendritic plateau potentials to be generated in the context of the hippocampal neural circuit. Previous work has shown that (1) plateau potentials are positively regulated by excitatory inputs from entorhinal cortex (*Bittner et al., 2015*; *Takahashi and Magee, 2009*; *Milstein et al., 2015*), (2) they are negatively regulated by dendrite-targeting inhibition (*Grienberger et al., 2017*; *Milstein et al., 2015*; *Lovett-Barron et al., 2012*; *Royer et al., 2012*; *Palmer et al., 2012*), (3) they occur more frequently in novel environments (*Cohen et al., 2017*) and precede the emergence of new place fields (*Sheffield et al., 2017*), and (4) introduction of a fixed reward site induces large shifts in the place field locations of many place cells in a population, as assayed by calcium imaging (*Turi et al., 2019*). In order to explore the consequences of these regulatory mechanisms on memory storage by BTSP at the network level, we next constructed a network model of the CA1 microcircuit that incorporates these critical elements to regulate plateau initiation (*Figure 7A*) and implements the above-described weight-dependent model of BTSP (*Figures 5 and 6*) at each input to the network.

In a population of 500 firing rate model CA1 pyramidal neurons, plateaus were positively regulated by a long-range feedback input from entorhinal cortex and negatively regulated by local feedback inhibition (*Figure 7A and B*; *Stefanelli et al., 2016*). Generation of plateau potentials within the population of CA1 neurons in the model was stochastic, which would result from fluctuations in inputs from entorhinal cortex that occasionally cross a threshold for the generation of a plateau potential in different cells at different times. The presence of reward delivered at a fixed goal location was implemented as an increase in input from entorhinal cortex (*Boccara et al., 2019*; *Butler et al., 2019*), although an equivalent increase in plateau generation could result instead from neuromodulatory input that directly increased dendritic excitability or reduced dendritic inhibition (*Sjöström et al., 2008*; *Pi et al., 2013*; *Tyan et al., 2014*; *Guerguiev et al., 2017*).

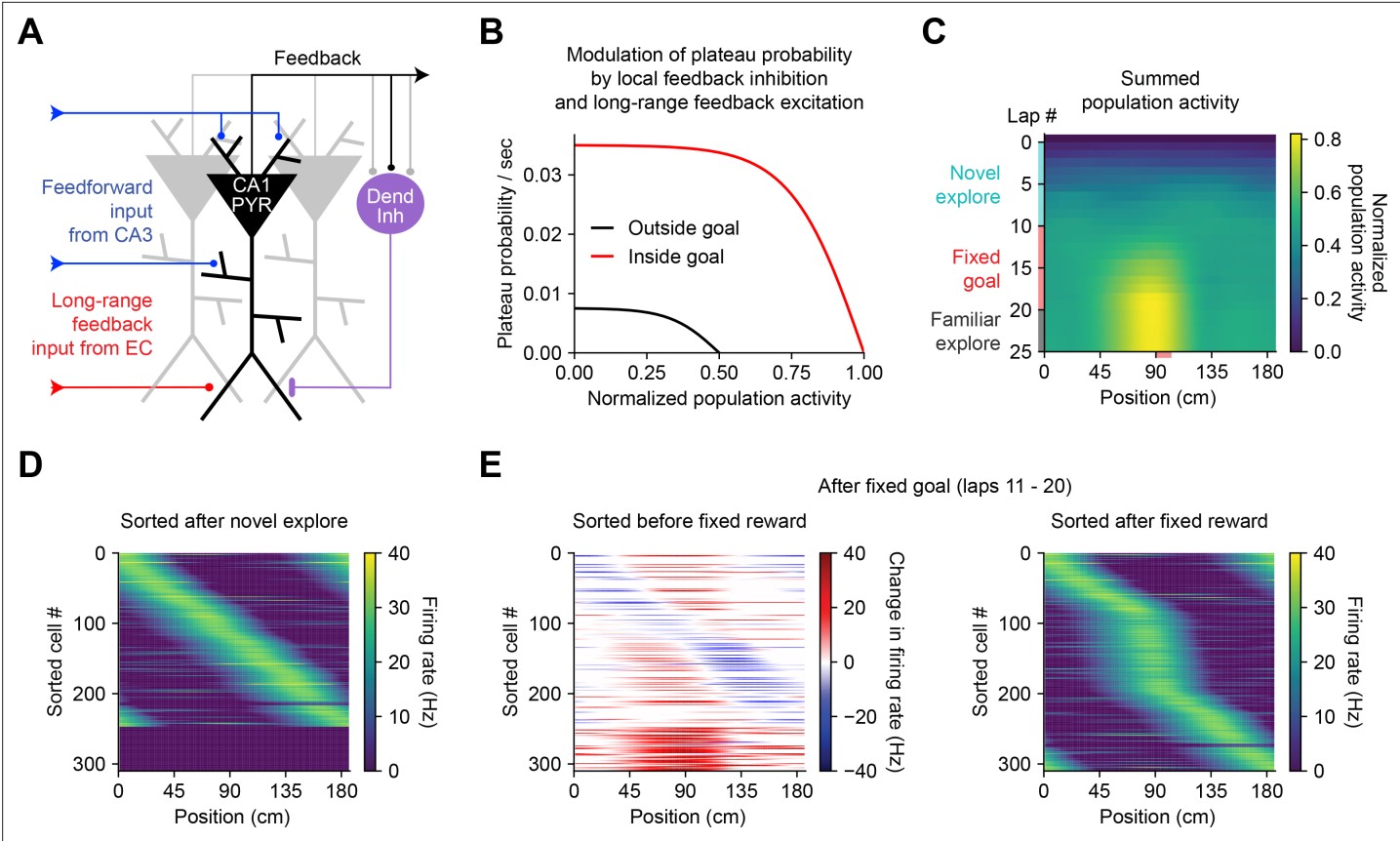

**Figure 7.** Bidirectional behavioral timescale synaptic plasticity (BTSP) enables rapid adaptation of population representations in a network model. (**A**) Diagram depicts components of a hippocampal network model. A population of CA1 pyramidal neurons receives spatially tuned excitatory input from a population of CA3 place cells and a long-range feedback input from entorhinal cortex (EC) that signals the presence of a behavioral goal. The output of CA1 pyramidal neurons recruits local feedback inhibition from a population of interneurons. (**B**) The probability that model CA1 neurons emit plateau potentials and induce bidirectional plasticity is negatively modulated by feedback inhibition. As the total number of active CA1 neurons increases (labeled 'normalized population activity'), feedback inhibition increases, and plateau probability decreases until a target level of population activity is reached, after which no further plasticity can be induced (black). A long-range feedback input signaling the presence of a goal increases plateau probability, resulting in a higher target level of population activity inside the goal region (red). (**C**) Each row depicts the summed activity of the population of model CA1 pyramidal neurons across spatial positions during a lap of simulated running. Laps 1–10 reflect exploration of a previously unexplored circular track. During laps 11–20, a goal is added to the environment at a fixed location (90 cm). During laps 21–25, the goal is removed for additional exploration of the now familiar environment. (**D – E**) Activity of individual model CA1 pyramidal neurons during simulated exploration as described in (**C**). (**D**) The firing rates of model neurons are sorted by the peak location of their spatial activity following 10 laps of novel exploration. ~250 neurons have acquired place fields. A fraction of the population remains inactive and untuned. (**E**) Left: changes in firing rate of model neurons after 10 laps of goal-directed search shows place field acquisition and translocation. Right: the firing rates of model neurons are re-sorted by their new peak locations. An increased fraction of neurons express place fields near the goal position. The remaining silent ~200/500 neurons that did not acquire a place field are not shown.

The online version of this article includes the following figure supplement(s) for figure 7:

**Figure supplement 1.** New place field acquisition and pre-existing place field translocation in a network model of goal-directed navigation (related to *Figure 7*).

During goal-directed navigation, hippocampal neurons have been shown to preferentially acquire new place fields near behaviorally relevant locations, and to translocate existing place fields toward those locations (*Dupret et al., 2010*; *Zaremba et al., 2017*; *Turi et al., 2019*; *Hollup et al., 2001*; *Gauthier and Tank, 2018*; *Lee et al., 2020*). We modeled this situation by simulating a virtual animal running on a circular treadmill for three separate phases of exploration (*Figure 7C*). At each time step (10 ms), instantaneous plateau probabilities were computed for each cell (*Figure 7B*), determining which neurons would initiate a dendritic plateau and undergo plasticity. During the first few laps of simulated exploration, CA1 pyramidal neurons rapidly acquired place fields that, as a population,

uniformly tiled the track (*Figure 7C and D*). As neurons increased their activity over time, feedback inhibition increased proportionally and prevented further plasticity (*Figure 7A–C*). During the next phase a goal was presented at a fixed location, resulting in both acquisition of new place fields nearby the goal location in a population of initially silent neurons, and translocation of place fields toward the goal location in a separate population of cells with pre-existing fields (*Figure 7E*, left; *Figure 7—figure supplement 1*). Overall, this resulted in an increased proportion of place cells with fields near the goal position (*Figure 7E*, right), recapitulating experimentally observed modifications in CA1 network activity during goal-directed behavior (*Zaremba et al., 2017*). The asymmetric time course of BTSP caused the population representation of the goal in the model to peak before the goal location itself, producing a predictive memory representation of the path leading to the goal (*Mehta et al., 1997*; *Stachenfeld et al., 2017*). Simulated place cell activity remained stable in a final phase of exploration without reward (*Figure 7C* and *Figure 7—figure supplement 1*). These network modeling results demonstrate that plasticity regulated by local network activity and long-range feedback, rather than by pairwise correlations, can enable populations of place cells to rapidly adapt their spatial representations to changes in the environment without any compromise in selectivity.

## Discussion

In summary, we observed translocation of hippocampal place fields by dendritic plateau potentials and characterized the underlying synaptic learning rule. We found that BTSP is bidirectional, inducing both synaptic potentiation and synaptic depression in neurons expressing pre-existing place fields. The direction of plasticity is determined by the synaptic weight of each excitatory input prior to a plateau potential, and the time interval between synaptic activity and a plateau. The large magnitude of synaptic weight changes enables BTSP to rapidly reshape place field activity in a small number of trials. These results corroborate recent work showing that changes in place field firing in CA1 could be induced by juxtacellular current injection, which was correlated with the occurrence of long duration complex spikes (*Diamantaki et al., 2018*). Here, we used intracellular stimulation and recording to reliably evoke dendritic calcium spikes with precise timing and duration, and to monitor subthreshold changes in $V_m$ dynamics, which enabled inference of the underlying synaptic learning rule.

The time and synaptic weight dependence of BTSP suggests that it is driven by an input-specific process rather than nonselective heterosynaptic (*Lynch et al., 1977*) or homeostatic plasticity (*Mendez et al., 2018*; *Hengen et al., 2016*), or modulation of cellular excitability (*Chandra and Barkai, 2018*; *Titley et al., 2017*). A significant role for changes in inhibitory synaptic weights is unlikely given that (1) inhibitory neurons in CA1 exhibit low levels of spatial selectivity (*Grienberger et al., 2017*), (2) homosynaptic potentiation of excitatory inputs by dendritic plateau potentials can be induced with GABAergic inhibition blocked (*Bittner et al., 2017*), and (3) inhibitory input to CA1 neurons does not change following induction of synaptic potentiation by BTSP (*Grienberger et al., 2017*).

The voltage perturbation experiments we performed (*Figure 4*) showed that BTSP does not depend on the activation state of the postsynaptic neuron. These results point to a fundamental difference between BTSP and existing Hebbian models of plasticity. In most previous models, including the aforementioned 'three-factor' plasticity models, the firing rate (*He et al., 2015*; *Brzosko et al., 2015*; *Markram et al., 1997*; *Bi and Poo, 1998*), or sustained level of global depolarization (*Clopath et al., 2010*; *Artola et al., 1990*; *Brandalise and Gerber, 2014*) at the time of presynaptic spiking primarily determines whether a synaptic weight increases or decreases (*Gerstner et al., 2018*; *Abbott and Nelson, 2000*; *Caporale and Dan, 2008*). Our voltage perturbation experiments (*Figure 4* and *Figure 4—figure supplement 2*) show that the direction of plasticity is not determined by either global depolarization or spiking.

This lack of dependence on the postsynaptic activity or output could enable plasticity to be robust to fluctuations in postsynaptic state due to noise or network oscillations (e.g. theta or gamma) (*Buzsáki and Moser, 2013*), and may allow the postsynaptic state to subserve other functions, such as temporal coding, without interfering with ongoing synaptic weight modifications. Furthermore, while in traditional Hebbian models of plasticity, short timescale synchrony between pre- and postsynaptic activity modifies weights to reinforce pre-existing correlations, BTSP instead provides a mechanism to either create new pairwise activity correlations 'from scratch', or remove pre-existing ones based on delayed outcomes. Our network model (*Figure 7*) highlights how this fundamental element of BTSP could shape spatial memory storage at the network level, allowing neuronal circuits to rapidly acquire

population-level representations of previously unencountered environmental features, and to modify outdated representations. This model also demonstrated that, if plateau potentials are generated by a mismatch between local circuit output and target information relayed by long-range feedback, BTSP can implement objective-based learning (*Richards et al., 2019b*; *Sacramento, 2018*; *Payeur et al., 2020*).

Together our experimental and modeling results establish BTSP as a potent mechanism for rapid and reversible learning. In addition to providing insight into the fundamental mechanisms of spatial memory formation in the hippocampus, these findings suggest new directions for general theories of biological learning and the development of artificial learning systems (*Guerguiev et al., 2017*; *Payeur et al., 2020*; *Bono and Clopath, 2017*; *Richards and Lillicrap, 2019a*; *Lillicrap et al., 2020*).

## Materials and methods
### Animals and procedures

All experimental methods were approved by the Janelia or Baylor College of Medicine Institutional Animal Care and Use Committees (Protocols 12–84 and 15–126). All experimental procedures in this study, including animal surgeries, behavioral training, treadmill and rig configuration, and intracellular recordings, were performed identically to a previous detailed report (*Bittner et al., 2017*) in an overlapping set of experiments, and are briefly summarized here.

In vivo experiments were performed in 6- to 12-week-old mice of either sex. Craniotomies above the dorsal hippocampus for simultaneous whole-cell patch clamp and local field potential (LFP) recordings, as well as affixation of head bar implants were performed under deep anesthesia. Following a week of recovery, animals were prepared for behavioral training with water restriction, handling by the experimenter, and addition of running wheels to their home cages. Mice were trained to run on the cue-enriched linear treadmill for a dilute sucrose reward delivered through a licking port once per lap (~187 cm). A MATLAB GUI interfaced with a custom microprocessor-controlled system for position-dependent reward delivery and intracellular current injection. Animal-run velocity was measured by an encoder attached to one of the wheel axles.

Plasticity was induced in vivo by injecting current (700 pA, 300 ms) intracellularly into recorded CA1 neurons to evoke dendritic plateau potentials at the same position on the circular treadmill for multiple consecutive laps. In most cases, plateaus were evoked on five consecutive laps (*Figure 1—figure supplement 1E*, left). However, during some experiments, large changes in the spatial $V_m$ ramp depolarization could be observed to develop after as few as one plateau (consistent with the observation that plasticity could be induced by a single spontaneously-occurring plateau), and so fewer induction laps were used. In other experiments, plateaus were induced on more than five consecutive laps if place field expression remained weak after the first five trials (*Figure 1—figure supplement 1E*, left). The source of this variability across cells/animals is not yet clear, and requires future investigation. Overall, this procedure induced changes in spatial $V_m$ ramp depolarization in 100% of cells in which it was attempted by three investigators. In some cells, the initial place field was first induced by this procedure, and then the procedure was repeated a second or third time in the same cell with plateaus induced at different locations. In those cases, there was no systematic difference in the number of plateaus required to induce the first place field compared to subsequent fields (*Figure 1—figure supplement 1E*, right).

Since the time window for plasticity induction by BTSP extends for seconds around each plateau, and plateaus were typically evoked on multiple consecutive laps, the changes in synaptic weights induced by BTSP depended on the run behavior of the animals across all induction laps. We showed in *Figure 3D* that the spatial width of place fields induced by BTSP varied with the average velocity of animals across all plasticity induction laps. Another factor that contributed to the spatial width of induced fields is the proximity of the evoked plateaus to the reward site, as animals tended to stop running briefly to lick near the fixed reward site. Variability across laps in either the run velocity or the duration of pauses could pose a challenge in trying to relate spatial changes in $V_m$ ramp depolarization to the time delay to the plateau (see below). *Figure 1—figure supplement 1* shows the full run trajectories of animals during all plasticity induction laps for the five representative example cells shown in *Figure 1*. While some variability across induction laps was observed, each animal tended to run consistently at similar velocities across laps.

## In vivo intracellular electrophysiology

To establish whole-cell recordings from CA1 pyramidal neurons, an extracellular LFP electrode was lowered into the dorsal hippocampus using a micromanipulator until prominent theta-modulated spiking and increased ripple amplitude was detected. Then a glass intracellular recording pipette was lowered to the same depth while applying positive pressure. The intracellular solution contained (in mM): 134 K-gluconate, 6 KCl, 10 HEPES, 4 NaCl, 0.3 MgGTP, 4 MgATP, 14 Tris-phosphocreatine, and in some recordings, 0.2% biocytin. Current-clamp recordings of intracellular membrane potential ($V_m$) were amplified and digitized at 20 kHz, without correction for liquid junction potential. The silent-cell population of neurons ($n$ = 29) contained recordings from 17 neurons that have been previously reported (*Bittner et al., 2017*).

In a subset of experiments (*Figure 4* and *Figure 4—figure supplement 2*), in addition to position-dependent step current to evoke plateau potentials, additional current was injected either to depolarize neurons beyond spike threshold or to hyperpolarize neurons below spike threshold, during plasticity induction laps. While these perturbations to $V_m$ at the soma are expected to attenuate along the path to distal dendrites (*Golding et al., 2005*), the pairing of back-propagating action potentials with synaptic inputs has been shown to significantly amplify dendritic depolarization (*Jarsky et al., 2005*; *Stuart and Häusser, 2001*; *Migliore et al., 1999*; *Schiller and Schiller, 2001*). Simulations of a biophysically detailed CA1 place cell model with realistic morphology and distributions of dendritic ion channels (*Grienberger et al., 2017*) suggest that somatic depolarization of a silent CA1 cell increases distal dendritic depolarization, and that somatic hyperpolarization of a place cell substantially reduces distal dendritic depolarization at the peak of its place field (*Figure 4—figure supplement 1*).

## Place field analysis

To analyze subthreshold $V_m$ ramps, action potentials were first removed from raw $V_m$ traces and linearly interpolated, then the resulting traces were low-pass filtered (<3 Hz). For each of 100 equally sized spatial bins (~1.85 cm), $V_m$ ramp amplitudes were computed by averaging across 10 laps of running on the treadmill both before and after plasticity induction. The spatially binned ramp traces were then smoothed with a Savitzky-Golay filter with wrap-around. Ramp amplitude was quantified as the difference between the peak and the baseline (average of the 10% most hyperpolarized bins). For cells with a second place field induced, the same baseline $V_m$ value determined from the period before the second induction was also used to quantify ramp amplitude after the second induction. Plateau duration was estimated as the duration of intracellular step current injections, or as the full width at half maximum $V_m$ in the case of spontaneous naturally occurring plateaus.

$V_m$ ramp half-width (*Figure 2D* and *Figure 6—figure supplement 1C*) was calculated from the $\Delta V_m$ traces as the time (s) or distance (cm) between the plateau and the final return of $\Delta V_m$ to zero (or at least to 25% of min; see *Figure 2—figure supplement 1G*). In most cases this only occurred on one side of the plateau, during either the running period before or after the plateau. In 5/26 inductions, the mouse ran so quickly that the $\Delta V_m$ did not have time to reach 25% of min on either side of the plateau (*Figure 2—figure supplement 1G*), resulting in an underestimation of the ramp half-width. The average velocity was calculated as the mean velocity of the mouse from the plateau to the end of the plasticity (*Figure 2—figure supplement 1*).

In order to relate spatial changes in $V_m$ ramp depolarization to the time delay to a plateau (e.g. *Figures 2E, F, 3A–F, I, 4B, C, E, F and 6E*), we assigned to each spatial position the shortest time delay to plateau that occurred across multiple induction laps (*Figure 1—figure supplement 1*). This is a conservative estimate, as the shortest delay between presynaptic activity and postsynaptic plateau will generate the largest overlap between ET and IS, and will result in the largest changes in synaptic weight. While this method is imperfect and did discard variability in running behavior across laps, it enabled direct comparison of the time course of BTSP across neurons. We also note that, to generate the modeling results shown in *Figure 6*, the full run trajectory of each animal during all induction laps, including pauses, was provided as input to the model (see details below). This resulted in good quantitative agreement between experimentally recorded and modeled spatial $V_m$ ramps (*Figure 6D*). Since not all possible pairs of initial ramp amplitude and time delay relative to plateau onset were sampled in the experimental dataset, expected changes in ramp amplitude (e.g. *Figure 3I*) were predicted from the sampled experimental or model data points by a two-dimensional Gaussian process regression and interpolation procedure using a rational quadratic

covariance function, implemented in the open-source Python package sklearn (**Abraham et al., 2014**; **Rasmussen and Williams, 2006**).

To statistically compare $\Delta V_m$ vs. time plots among groups each individual induction trace was binned in time (average of values in 80, 100 ms, bins from –4 to +4 s). The number of points in each bin for each group is as follows: silent cells (–4, + 4 s): $n$ = 19, 19, 19, 0, 20, 20, 21, 21, 25, 26, 26, 27, 27, 27, 27, 27, 28, 28, 28, 28, 29, 29, 29, 29, 29, 29, 29, 29, 29, 29, 29, 29, 29, 29, 29, 29, 29, 29, 29, 29, 29, 29, 29, 29, 29, 29, 29, 29, 29, 27, 25, 25, 25, 24, 21, 20, 19, 17, 16, 14, 14, 10, 9, 9, 8, 8, 7, 7, 7, 7, 7, 6, 6, 6, 6, 6. Silent + depolarization (–4, + 4 s): $n$ = 2, 2, 4, 5, 5, 6, 6, 6, 6, 6, 6, 7, 7, 7, 8, 8, 8, 8, 8, 8, 8, 8, 8, 8, 8, 8, 8, 8, 8, 8, 8, 8, 8, 8, 8, 8, 8, 8, 8, 8, 8, 8, 8, 8, 8, 8, 8, 8, 8, 8, 8, 8, 8, 8, 8, 8, 8, 8, 7, 7, 6, 6, 6, 6, 5, 5, 5, 5, 3, 3. Depolarized PCs (–4 to +4 s): $n$ = 6, 6, 6, 6, 6, 5, 5, 5, 4, 4, 5, 6, 6, 6, 7, 7, 6, 6, 6, 7, 7, 8, 7, 7, 7, 7, 6, 6, 5, 5, 5, 4, 4, 4, 4, 4, 4, 4, 4, 4, 4, 5, 5, 5, 5, 5, 5, 5, 6, 7, 8, 8, 8, 9, 10, 14, 14, 15, 15, 15, 15, 15, 14, 14, 14, 14, 14, 14, 14, 14, 14, 14, 14, 14, 13, 12, 12, 10, 9, 9. All PCs (–1 to +4 s): $n$ = 26, 26, 26, 26, 26, 26, 26, 26, 26, 26, 26, 26, 26, 26, 26, 26, 26, 26, 26, 26, 26, 26, 26, 26, 26, 26, 26, 26, 26, 26, 26, 26, 26, 26, 26, 26, 26, 26, 26, 26, 26, 26, 26, 25, 24, 24. PCs + hyperpolarization (–1 to +4 s): $n$ = 5, 6, 6, 7, 7, 8, 8, 8, 8, 8, 8, 8, 8, 8, 8, 8, 8, 8, 8, 8, 8, 8, 8, 8, 8, 8, 8, 8, 8, 8, 8, 8, 8, 8, 8, 8, 8, 8, 8, 8, 8, 8, 8, 8, 8, 8, 8, 8, 8, 8, 8, 8, 8, 8, 8, 8, 8, 8, 8, 8. Silent+ large hyperpolarization (–4 to +4): $n$ = 4, 4, 4, 4, 5, 5, 5, 6, 6, 6, 6, 6, 6, 7, 7, 7, 7, 7, 7, 7, 7, 7, 7, 7, 7, 7, 7, 7, 7, 7, 7, 7, 7, 7, 7, 7, 7, 7, 7, 7, 7, 7, 7, 7, 7, 7, 7, 6, 6, 6, 6, 5, 5, 5, 5, 4, 4, 4, 4, 4, 4, 3, 3, 3, 3, 2, 2, 2, 2, 2, 2, 2, 2, 2, 2.

## Quantification and statistical analysis

Statistical details of experiments can be found in the figure legends. Unless otherwise specified, measured values and ranges reflect mean ± SEM. Significance was defined as p < 0.05. Sample sizes were not determined by statistical methods, but efforts were made to collect as many samples as was technically feasible. No data or subjects were excluded from any analysis.

## Computational modeling

### Weight-dependent BTSP model

In **Figures 5 and 6**, we provide a mathematical model of the synaptic learning rule underlying bidirectional BTSP. In this 'weight-dependent' model, the direction and magnitude of plasticity at excitatory synapses from spatially tuned CA3 place cell inputs onto a CA1 pyramidal cell are determined by (1) the timing of presynaptic spiking relative to postsynaptic plateau potentials and (2) the current weight of each synapse just prior to a plateau. While in **Figure 5**, discrete spikes were provided as presynaptic inputs to the model, in **Figure 6**, presynaptic inputs were provided as continuous firing rates. This model contained nine free parameters (described in detail below), which were fit to the experimental data using an iterative, bounded, stochastic search procedure based on the simulated annealing algorithm (**Milstein, 2021a**; **Milstein, 2021b**). This optimization sought to minimize the difference between the experimentally recorded place cell $V_m$ ramp depolarizations (**Figures 1–3**) and those predicted by the model (**Figure 6D and E**). Parameter optimization was considered to converge after sampling 30,000 distinct model configurations. Below we describe the model formulation in detail.

A CA1 place cell was modeled as receiving excitatory input from a population of 200 CA3 place cells with spatially tuned firing fields spaced uniformly across an ~185 cm circular track (**Figure 3J**). The firing rate $R_i$ of an individual input with place field at position $y_i$ depended on the recorded run trajectory of the animal $x\left(t\right)$ (**Figure 6A**, first and second rows):

$$R_i\left(t\right) = R_{max} * e^{-\frac{1}{2}\left(\frac{y_i - x(t)}{\sigma}\right)^2} \tag{6}$$

where $R_{max}$ is a maximum firing rate of 40 Hz at the peak of a place field, and $\sigma$ determines the width of the place field. $\sigma$ was set such that CA3 place field inputs had a full floor width ($6 * \sigma$) of 90 cm (half-width of ~34 cm) (**Mizuseki et al., 2012**), though models tuned with alternative values of $\sigma$ generated quantitatively similar predictions ('60 cm input field widths' in **Figure 6—figure supplement 2**). The complete run trajectory of each animal during consecutive plasticity induction laps, including pauses in running between laps, was provided as a continuous input to the model. In accordance with

experimental data (**Bittner et al., 2015**; **Grienberger et al., 2017**), the firing rates of model place cell inputs were set to zero during periods when the animal stopped running.

The $V_m$ ramp depolarization of a CA1 place cell as a function of position, $V(x)$, was modeled as a weighted sum of the spatial firing rates of the CA3 place cell inputs. We assumed that in silent cells prior to plasticity induction, all inputs had an initial synaptic weight of 1. This produced a background level of depolarization, $V_b$, which was subtracted from the total weighted sum to calculate the ramp amplitude (**Figure 6C–E**):

$$V(x) = c * \sum_i W_i * R_i(x) - V_b \tag{7}$$

The scaling factor $c$ was calibrated such that if the synaptic weights of CA3 place cell inputs varied between 1 and 2.5 as a Gaussian function of their place field locations, the postsynaptic CA1 cell would express a $V_m$ ramp with 108 cm width and 6 mV peak amplitude, consistent with previous measurements of place field properties and the degree of synaptic potentiation by BTSP (**Bittner et al., 2017**). For CA1 place cells already expressing a place field before plateaus were evoked at a second location, the initial synaptic weights were estimated by using least squares approximation to fit the experimentally recorded initial $V_m$ ramp.

At each input , a postsynaptic eligibility trace $ET_i$ was activated by presynaptic firing $R_i$ and decayed with a seconds-long time course $\tau_{ET}$ (**Figure 6A**, third row):

$$\tau_{ET} * \frac{dET_i}{dt} = -ET_i + \lambda_{ET} * R_i \tag{8}$$

The scaling factor $\lambda_{ET}$ was chosen such that the maximum amplitude of $ET$ does not exceed 1. For single spike inputs, as shown in **Figure 5**, the firing rate $R_i$ was replaced with a delta function $\delta(t - t^s)$ where $t^s$ is the time of the spike.

Postsynaptic dendritic plateau potentials during each induction lap $\mu$ with onset at time $t^p$ and duration $d$ activated an instructive signal $IS$ that was broadcast to all synapses and decayed exponentially with time course $\tau_{IS}$ (**Figure 6A**, fourth row):

$$\tau_{IS} * \frac{dIS}{dt} = -IS + \lambda_{IS} * P(t^p, d) \tag{9}$$

where $P$ is a binary function that takes a value of 1 during a plateau and 0 otherwise. The scaling factor $\lambda_{IS}$ was chosen such that the maximum amplitude of $IS$ does not exceed 1. The duration of experimentally induced plateaus were typically 300 ms, but spontaneous plateaus were recorded with duration up to ~800 ms.

Next, temporal overlap of eligibility traces $ET_i$ and instructive signals $IS$ (**Figure 6A**, fifth row) were considered to drive saturable potentiation and depression processes independently at each synapse. The sensitivity of these two processes $q^+$ and $q^-$ to the amplitude of plasticity signal overlap was defined by generalized sigmoid functions $s(x, \alpha, \beta)$ with a scale and offset to meet the following edge constraints: $s = 0$ when $x = 0$, $s = 1$ when $x = 1$:

$$\hat{s}(x, \alpha, \beta) = \frac{1}{1 + e^{(-\beta(x-\alpha))}} \tag{10}$$

$$s(x, \alpha, \beta) = \frac{\hat{s}(x, \alpha, \beta) - \hat{s}(0, \alpha, \beta)}{\hat{s}(1, \alpha, \beta) - \hat{s}(0, \alpha, \beta)} \tag{11}$$

$$q^+(ET_i * IS) = s(ET_i * IS, \alpha^+, \beta^+) \tag{12}$$

$$q^-(ET_i * IS) = s(ET_i * IS, \alpha^-, \beta^-) \tag{13}$$

where $\alpha^\pm$ and $\beta^\pm$ control the threshold and slope of the sigmoidal gain functions for potentiation and depression.

Finally, to capture the dependency of changes in synaptic weight $\frac{dW_i(t)}{dt}$ on the current value of synaptic weight $W_i$ at each input during plasticity induction, we chose a two-state non-stationary kinetic model of the following form:

$$\overset{Inactive}{I} \underset{k^- * q^-(ET_i * IS)}{\overset{k^+ * q^+(ET_i * IS)}{\rightleftharpoons}} \overset{Active}{A} \tag{14}$$

According to this formulation, independent and finite synaptic resources at each synapse occupied either an inactive state $I$ or an active state $A$, and transitioned between states with rates controlled by the constants $k^{\pm}$ and the gain functions $q^{\pm}$ described above. The synaptic weight of each input $W_i$ was defined as proportional to the occupancy of the active state $A$:

$$W_i = A * W_{max} \tag{15}$$

where $0 \leq A \leq 1$, and $W_{max}$ is a free parameter controlling the maximum value of synaptic weight. Since the occupancy of each state in a kinetic model constrains the flow of finite resources between states, the net change in synaptic weight $\frac{dW_i}{dt}$ at each input naturally depended on the current value of synaptic weight $W_i$ :

$$\frac{dW_i}{dt} = \left(W_{max} - W_i\right) * k^+ * q^+ \left(ET_i * IS\right) - W_i * k^+ * q^- \left(ET_i * IS\right) \tag{16}$$

Changes in synaptic weight $W_i$ were calculated by integrating the net rate of change of synaptic weight $\frac{dW_i}{dt}$ over the duration of plasticity induction. In practice, for simplicity and efficiency of computation during parameter optimization, we numerically approximated $W_i$ by holding the value of $W_i$ constant for the duration of each induction lap, and updating $W_i$ once at the end of each induction lap (*Figure 6*). Equivalent results were obtained by updating $W_i$ continuously in 10 ms steps without requiring any change in parameters.

The weight-dependent model of the BTSP rule contained nine free parameters. The range of parameter values that fit the experimental data ($n$ = 26 plasticity inductions in 24 neurons with pre-existing place fields) were as follows (mean ± SEM): (1) $\tau_{ET}$: 863.91 ± 113.93 ms, (2) $\tau_{IS}$: 542.76 ± 95.47 ms, (3) $\alpha^+$: 0.24 ± 0.05, (4) $\beta^+$: 30.32 ± 6.50, (5) $\alpha^-$: 0.09 ± 0.04, (6) $\beta^-$: 2260.61 ± 1529.97, (7) $k^+$: 2.27 ± 0.49/ s, (8) $k^-$ : 0.33 ± 0.11/ s, (9) $W_{max}$: 4.02 ± 0.17. The results of the model in response to simpler single-spike inputs in *Figure 5A–C* were obtained with the following parameter values: (1) $\tau_{ET}$: 2500 ms, (2) $\tau_{IS}$: 1500 ms, (3) $\alpha^+$: 0.5, (4) $\beta^+$: 4, (5) $\alpha^-$: 0.01, (6) $\beta^-$: 44.44, (7) $k^+$: 1.7/ s, (8) $k^-$: 0.204/s, (9) $W_{max}$: 5.

## Alternative formulations of the weight-dependent BTSP model

Given the complexity of the above model, we also tested a number of alternative formulations to determine if the experimental data could be accounted for by a simpler model. First, we tested whether the filter time constants $_{ET}$ and $_{IS}$ that control the duration of the $ET$ and $IS$ could be shorter by constraining their values during parameter optimization to be less than 50 ms. This model variant performed poorly in predicting the depression component of BTSP ('short timescale ET and IS' in *Figure 5E* and *Figure 6—figure supplement 2A and B*). This supports the notion that intermediate signals with durations longer than either voltage or calcium are required for the long timescale of BTSP. This also demonstrates that the nonlinear gain functions $q^{\pm}$ are not able to compensate for shorter duration $ET$ or $IS$.

Next, we determined whether the nonlinear gain functions $q^{\pm}$ could instead be linear by replacing both the sigmoidal $q^+$ and $q^-$ with the identity function:

$$q^+ \left(ET_i * IS\right) = ET_i * IS \tag{17}$$

$$q^- \left(ET_i * IS\right) = ET_i * IS \tag{18}$$

This model variant also failed to account for synaptic depression by BTSP ('linear $q^+$ and $q^-$' in *Figure 5D* and *Figure 6—figure supplement 2A and B*), suggesting that nonlinearity of bidirectional plasticity is an important feature of the weight-dependent BTSP model.

## Goal-directed spatial learning model

To investigate the implications of bidirectional BTSP for spatial learning by a population of CA1 place cells (*Figure 7*), we constructed a network model comprised of 500 CA1 pyramidal cells each receiving input from a population of 200 CA3 place cells with place fields spaced at regular intervals spanning the ~185 cm circular track. The synaptic weights at inputs from model CA3 place cells to model CA1 cells were controlled by the weight-dependent model described above (*Figures 5 and 6*). For this purpose, the nine free parameters of the model were calibrated to match synthetic target

$V_m$ ramp data as follows: (1) lap running was simulated at a constant run velocity of 25 cm/s, (2) in an initially silent cell, plasticity was induced by three consecutive laps with one 300 ms long plateau per lap evoked at a fixed location, (3) after plasticity, the induced place field $V_m$ ramp had an asymmetric shape (~75 cm rise, ~ 35 cm decay) and a peak amplitude of 8 mV, (4) three additional plasticity induction laps with plateaus evoked at a location 3 s behind the peak location of the initial place field resulted in a 5 mV decrease in ramp amplitude at the initial peak location, and an 8 mV peak ramp amplitude at the new translocated peak position.

Before simulated exploration, all synaptic weights were initialized to a value of 1, which resulted in zero ramp depolarization in all model CA1 cells. Under these baseline conditions, each model CA1 neuron $k$ had a probability $p_k(t) = p^{basal} = 0.0075$ of emitting a single dendritic plateau potential in 1 s of running. During each 10 ms time step, this instantaneous probability $p_k(t)$ was used to weight biased coin flips to determine which cells would emit a plateau. This stochasticity can be thought of as reflecting fluctuations in the synaptic input arriving to each cell from the long-range cortical input pathway that occasionally drives the neuron to cross a threshold for generation of a dendritic calcium spike. If a cell emitted a plateau, it persisted for a fixed duration of 300 ms and was followed by a 500 ms refractory period during which $p_k(t)$ was transiently set to zero.

After the first lap, CA1 neurons that had emitted at least one plateau and had induced synaptic potentiation produced nonzero ramp depolarizations (*Figure 7C*). The output firing rates $R^{CA1}_{\mu,k}$ of each CA1 neuron $k$ on lap $\mu$ were considered to be proportional to their ramp depolarizations $V_{\mu,k}(t)$ after subtracting a threshold depolarization of 2 mV. The activity $R^{INH}_\mu(t)$ of a single inhibitory feedback element was set to be a normalized sum of the activity of the entire population of CA1 pyramidal neurons:

$$R^{INH}_\mu(t) = \lambda * \sum_k R^{CA1}_{\mu,k}(t) \tag{19}$$

where the normalization constant $\lambda$ was chosen such that the activity of the inhibitory feedback neuron would be one if every CA1 pyramidal neuron expressed a single place field and as a population their place field peak locations uniformly tiled the track. Then, the probability that any CA1 neuron $k$ would emit a plateau $p_k(t)$ was negatively regulated by the inhibitory feedback term $R^{INH}_\mu(t)$ :

$$p_k(t) = \begin{cases} s\left(R^{INH}_\mu(t), \alpha^{basal}, \beta^{basal}\right) & R^{INH}_\mu(t) < \alpha^{basal} \\ 0 & R^{INH}_\mu(t) \geq \alpha^{basal} \end{cases} \tag{20}$$

where $\alpha^{basal}$ defined a target normalized population activity (set to 0.5) and $\beta^{basal}$ defined the slope of a descending sigmoid function with a maximum value of 0.0075 (*Figure 7B*).

In some laps, a specific location was assigned as the target of a goal-directed search. To mimic an increase in the activity of the long-range input from entorhinal cortex signaling the presence of the goal, the probability that a CA1 neuron would emit a plateau potential $p_k(t)$ was transiently increased when the simulated animal crossed the goal location for a period of 500 ms. Within the goal region, the relationship between $p_k(t)$ and $R^{INH}_\mu(t)$ was instead:

$$p_k(t) = \begin{cases} s\left(R^{INH}_\mu(t), \alpha^{goal}, \beta^{goal}\right) & R^{INH}_\mu(t) < \alpha^{goal} \\ 0 & R^{INH}_\mu(t) \geq \alpha^{goal} \end{cases} \tag{21}$$

where $\alpha^{goal}$ is an elevated target normalized population activity (set to 1.0) and $\beta^{goal}$ defines the slope of a descending sigmoid function with a maximum value of 0.035, corresponding to an elevated peak plateau probability (*Figure 7B*).

## Data and code availability

The complete dataset and Python code for data analysis and model simulation is available at https://github.com/neurosutras/BTSP (*Milstein, 2021c* copy archived at swh:1:rev:952cbb453ae80b2efe52f2936baa03e3a4689dc5).

## Acknowledgements

We are grateful to Karel Svoboda and Wulfram Gerstner for discussions, Nicolas Brunel and Nelson Spruston for comments on the manuscript, Grace Ng for contributing to software development, Roy Phillips for behavioral device development, Ivan Raikov for technical assistance with high-performance computing, and Kristopher Bouchard at LBNL for sharing large-scale computing resources provided by the National Energy Research Scientific Computing Center, a Department of Energy Office of Science User Facility (DE-AC02-05CH11231). This work was also made possible by computing allotments from NSF (XSEDE Comet, NCSA Blue Waters, and TACC Frontera) and supported by NIH BRAIN grant U19NS104590 and NIMH grant R01MH121979. SR and JM are supported by the Howard Hughes Medical Institute.

## Additional information

### Funding

| Funder | Grant reference number | Author |
| --- | --- | --- |
| National Institutes of Health | U19NS104590 | Aaron D Milstein Ivan Soltesz |
| National Institute of Mental Health | R01MH121979 | Aaron D Milstein |

The funders had no role in study design, data collection and interpretation, or the decision to submit the work for publication.

### Author contributions

Aaron D Milstein, Conceptualization, Formal analysis, Investigation, Methodology, Software, Visualization, Writing – original draft, Writing – review and editing; Yiding Li, Investigation, Methodology; Katie C Bittner, Christine Grienberger, Conceptualization, Investigation, Methodology, Visualization; Ivan Soltesz, Funding acquisition, Supervision, Writing – review and editing; Jeffrey C Magee, Conceptualization, Funding acquisition, Investigation, Supervision, Visualization, Writing – original draft, Writing – review and editing; Sandro Romani, Conceptualization, Funding acquisition, Investigation, Methodology, Supervision, Writing – original draft, Writing – review and editing

### Author ORCIDs

Aaron D Milstein http://orcid.org/0000-0002-7186-5779
Sandro Romani http://orcid.org/0000-0002-4727-4207

### Ethics

All experimental methods were approved by the Janelia or Baylor College of Medicine Institutional Animal Care and Use Committees (Protocol 12-84 & 15-126).

### Decision letter and Author response

Decision letter https://doi.org/10.7554/eLife.73046.sa1
Author response https://doi.org/10.7554/eLife.73046.sa2

## Additional files

### Supplementary files

• Transparent reporting form

### Data availability

The complete dataset, Python code for data analysis and model simulation, and additional MATLAB and Igor analysis scripts are available at https://github.com/neurosutras/BTSP (copy archived at swh:1:rev:952cbb453ae80b2efe52f2936baa03e3a4689dc5).

The following dataset was generated:

| Author(s) | Year | Dataset title | Dataset URL | Database and Identifier |
|---|---|---|---|---|
| Milstein AD, Li Y, Bittner KC, Grienberger C, Soltesz I, Magee JC, Romani S | 2021 | Bidirectional synaptic plasticity rapidly modifies hippocampal representations | https://github.com/neurosutras/BTSP | Github, neurosutras/BTSP |

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
