## [Editor Report]

This manuscript uses a combination of high-quality in vivo electrophysiology and modelling to demonstrate that Behavioural Time Scale Plasticity (BTSP) is bidirectional, and the amplitude and direction of this plasticity are dictated by the current weight of the inputs and not by the correlated activity of pairs of neurons. These findings challenge our current views on synaptic plasticity, which are primarily based on Hebb's concept. In addition, the network model used in this study demonstrates that this type of plasticity can rapidly reshape population activity to respond to environmental clues. This study will be of interest to the broad neuroscience audience and foster new ideas on biological and artificial learning.

---

## [Decision Letter]

**Decision letter after peer review:**

Thank you for submitting your article "Bidirectional synaptic plasticity rapidly modifies hippocampal representations" for consideration by *eLife*. Your article has been reviewed by 3 peer reviewers, and the evaluation has been overseen by a Reviewing Editor and Laura Colgin as the Senior Editor. The following individuals involved in review of your submission have agreed to reveal their identity: Jerome Epsztein (Reviewer #1); Richard Naud (Reviewer #2); Larry F Abbott (Reviewer #3).

Essential revisions:

The reviewers unanimously expressed their support for this manuscript. Their comments will help the authors to improve the clarity of the manuscript. The changes suggested by the reviewers do not require the authors to do additional experiments.

Reviewers' comments detail the points of improvement.

1. Please provide more details of the BTSP procedures and the behaviour of the mice.

2. Please elaborate on the rationale behind experiments depicted on Figure 2E, 3 and 4.

3. Please improve the clarity of data depicted on Figure 3D and 6A.

4. Please consider Rev.#2's comments on the approach of the computational section.

5. Please address questions raised about W and W_init_.

*Reviewer #2 (Recommendations for the authors):*

There are two parts to this article. One experimental part and one computational modelling part. My opinion of the experimental part is that it is very carefully done with impressive, compelling (and difficult!) experiments. The results are at first glance extremely perplexing, but the interpretation and follow-up experiments are clarifying a lot. In this respect, the manuscript is much improved over the biorxiv version I had read a year ago. My only note on the experimental part is that the p-value threshold for Figure 4 C and F used for denoting might have a bit of a higher rate of false positive because every time point would be considered as a comparison. This would be correctable by Bonferroni factors, but honestly, I don't think it is entirely necessary since there is a bit of ambiguity as to what hypothesis is being tested.

The computational part is compelling for the success of reproducing the data with a model that is intrinsically stable. I think the writing could be improved without too much work, focusing on two points.

The theory section was going deep in the implementation details. A level of depth that is uncommon for the main results part of a broad audience paper. I would suggest keeping the learning rule and maybe one or two crucial equations, but relegating many of the equations to the methods. For instance the linear ODE for the traces are so common that defining trace as a linear ODE or mono-exponential decay would be sufficient (if the equation remains in methods). Similar for the integration of the dW/dt over a trial. I want to stress that this is a stylistic point that I believe will help communicate the message, not something required for accepting the paper. Another stylistic point is the use of two-letter variables along with two conventions for multiplications (the Asterix and nothing as when tau multiples d λ /dt). I was also confused by the use of W_init_ in Equation 3, which in my view should be W (no init) all the time.

The results rely crucially less of the fact that q^+^ and q^-^ are nonlinear than on the fact that the nonlinearities are different. This should be clear from the fixed point analysis: consider Equation 8 with q^+^=q^-^ and you find that the fixed point is independent of timing even for nonlinear relationships. This can also be revealed when looking at the parameter values: the LTP term has a sigmoid with a small sensitivity (β^+^), a large threshold (α^+^) and a large saturating point (k^+^). The converse is true for the LTD term. Thus plotting the sigmoidal for each term should explain the Mexican hat: for a strong W, when the coincidence is very small, you only may only have a weak q^+^ effect (because the sensitivity difference). When the coincidence is medium, LTD dominates over LTP because q^-^ sharp threshold has been met without having reached the saturation of q^+^. When coincidence is strong q^+^ dominates because k^+^ is larger than k^-^ (and both sigmoids have saturated). All this depends on W of course. Personally, I would have liked to see q^+^ and q^-^ plotted, and how they combine to give the Mexican hat (or not). As in Figure 5B but showing q^+^ and q^-^ in addition. With respect to Figure 5B, the absence of potentiation at high coincidence IS*ET for the high W case is puzzling. Why is this not showing? W is not high enough? q^+^ should dominate in that regime.

Other notes. These are more comments which may or may not help rather than points to be addressed.

I would like to point that the plasticity model fits partially well with the model used in the recent paper by Payeur et al. 2021. In that paper, the bursts caused a potentiation that depended on previous potentiation (dependency on W). But that model had a very limited timing dependence and bursts would only lead to net LTD if they were not numerous enough relative to the number of spikes. Overall, a comparison with this model suggests that the BTSP model provided by Milstein et al. could be consistent with a coordination of plasticity, as acknowledged by the authors at the end of the discussion.

The model is also partially related with the model of Graupner,.… Brunel. Recently used to capture STDP phenomena by Aljadeff… Debanne. There, two threshold are present, one for LTP and one for LTD. So similar to the α^+^ and α^-^ being different. But the dependence in that model is very limited.

I thought the description of how W was kept between 1 and W_max_ was lacking. There are multiple ways of doing this. And the different types of implementation (hard threshold vs soft threshold) have been shown to lead to dramatically different functions in the STDP modelling literature.

I think there is a typo in the list of parameters given at the end as I don't find τ_IS_ and τ_ET_, but only τ_I_ and τ_E_.

*Reviewer #3 (Recommendations for the authors):*

This paper is beautifully done and well written. I have just a few suggestions:

1) It seems a pity that after elegantly formulating a model of BTSP, the authors final formulation of the synaptic change in Equations 3 and 4 is ill-defined. This is because the relationship between W and W_init_ is not specified. Wouldn't a proper formulation state that W_init_ = W, equation 3 does not describe dW/dt but rather a factor that determines ΔW through Equation 4, and that this latter process only takes place after a delay?

Alternatively, could W_init_ in the equation for dW/dt just be set to W? This would have the nice property of keeping W bounded between 0 and W_max_. Does this not work?

---

## [Author Response]

Essential revisions:The reviewers unanimously expressed their support for this manuscript. Their comments will help the authors to improve the clarity of the manuscript. The changes suggested by the reviewers do not require the authors to do additional experiments.Reviewers' comments detail the points of improvement.1. Please provide more details of the BTSP procedures and the behaviour of the mice.2. Please elaborate on the rationale behind experiments depicted on Figure 2E, 3 and 4.

The Reviewers raised the following important questions (paraphrased):

1) How variable were the number of plasticity induction trials used across animals, and in particular were different numbers of plateaus used for each induction when multiple inductions were performed in the same neuron?

2) How consistent was the running behavior of the mice during plasticity induction by experimentally-evoked dendritic plateau potentials? How does this affect analysis of the timescale of plasticity?

We have now included additional details in a new Figure 1—figure supplement 1, and have revised the Materials and methods section accordingly.

3. Please improve the clarity of data depicted on Figure 3D and 6A.

Please find our responses in line with the comments from Reviewer #1 in the posted Public Review.

4. Please consider Rev.#2's comments on the approach of the computational section.

Please find our responses in line with the comments from Reviewer #2 below.

5. Please address questions raised about W and W_init_.

Please find our responses in line with the comments from Reviewer #3 below.

Reviewer #2 (Recommendations for the authors):There are two parts to this article. One experimental part and one computational modelling part. My opinion of the experimental part is that it is very carefully done with impressive, compelling (and difficult!) experiments. The results are at first glance extremely perplexing, but the interpretation and follow-up experiments are clarifying a lot. In this respect, the manuscript is much improved over the biorxiv version I had read a year ago. My only note on the experimental part is that the p-value threshold for Figure 4 C and F used for denoting might have a bit of a higher rate of false positive because every time point would be considered as a comparison. This would be correctable by Bonferroni factors, but honestly, I don't think it is entirely necessary since there is a bit of ambiguity as to what hypothesis is being tested.

We agree that a false discovery rate correction (FDR) could be used here to adjust for repeated comparisons across time bins and groups. However, the hypothesis was that voltage-perturbations would change the sign of plasticity. Overall, hyperpolarization did not convert depression into potentiation, and depolarization did not convert potentiation into depression. Here we label temporal bins that showed a significant change in the amplitude (but not the direction) of plasticity. A high discovery rate is actually conservative in this case. We prefer to leave this analysis as such.

The computational part is compelling for the success of reproducing the data with a model that is intrinsically stable. I think the writing could be improved without too much work, focusing on two points.The theory section was going deep in the implementation details. A level of depth that is uncommon for the main results part of a broad audience paper. I would suggest keeping the learning rule and maybe one or two crucial equations, but relegating many of the equations to the methods. For instance the linear ODE for the traces are so common that defining trace as a linear ODE or mono-exponential decay would be sufficient (if the equation remains in methods). Similar for the integration of the dW/dt over a trial. I want to stress that this is a stylistic point that I believe will help communicate the message, not something required for accepting the paper.

The recommended changes have been made (P15 – P16).

Another stylistic point is the use of two-letter variables along with two conventions for multiplications (the Asterix and nothing as when tau multiples d λ /dt). I was also confused by the use of W_init_ in Equation 3, which in my view should be W (no init) all the time.

The recommended changes have been made (P16, L392). Please also see our response to the comments of Reviewer #3 below regarding the use of W_init_.

The results rely crucially less of the fact that q^+^ and q^-^ are nonlinear than on the fact that the nonlinearities are different. This should be clear from the fixed point analysis: consider Equation 8 with q^+^=q^-^ and you find that the fixed point is independent of timing even for nonlinear relationships. This can also be revealed when looking at the parameter values: the LTP term has a sigmoid with a small sensitivity (β^+^), a large threshold (α^+^) and a large saturating point (k^+^). The converse is true for the LTD term. Thus plotting the sigmoidal for each term should explain the Mexican hat: for a strong W, when the coincidence is very small, you only may only have a weak q^+^ effect (because the sensitivity difference). When the coincidence is medium, LTD dominates over LTP because q^-^ sharp threshold has been met without having reached the saturation of q^+^. When coincidence is strong q^+^ dominates because k^+^ is larger than k^-^ (and both sigmoids have saturated). All this depends on W of course. Personally, I would have liked to see q^+^ and q^-^ plotted, and how they combine to give the Mexican hat (or not).

To address this point, we have now added to Figure 5B a third example of dW/dt for intermediate synaptic weights which shows a Mexican hat sensitivity to signal overlap. We have now also added the following text to the Results section:

P16, L405: “If the depression process q^-^ has a lower threshold for activation than the potentiation process q^+^ (*61, 62*), the resulting change in synaptic weight dW/dt is positive and increases monotonically when initial weights are low, but is negative and non-monotonic when initial weights are high (Figure 5B). […] This is consistent with the in vivo data, which showed that negative changes in place field ramp *V*_m_ were largest at intermediate delays from a plateau (Figures 3B and 3I).”

As in Figure 5B but showing q^+^ and q^-^ in addition. With respect to Figure 5B, the absence of potentiation at high coincidence IS*ET for the high W case is puzzling. Why is this not showing? W is not high enough? q^+^ should dominate in that regime.

Consulting Equation 1 (P16, L392), when W is large and close to W_max_, the potentiation term vanishes, and there can be no potentiation, only depression, regardless of the amplitude of signal overlap. We now clarify this point with the following text:

P16, L394: “This formula can be obtained from a two-state model of finite synaptic resources (see Materials and methods). When the current synaptic weight W is near W_max_, the potentiation rate becomes zero, and when W is near zero, the depression rate becomes zero.”

Other notes. These are more comments which may or may not help rather than points to be addressed.I would like to point that the plasticity model fits partially well with the model used in the recent paper by Payeur et al. 2021. In that paper, the bursts caused a potentiation that depended on previous potentiation (dependency on W). But that model had a very limited timing dependence and bursts would only lead to net LTD if they were not numerous enough relative to the number of spikes. Overall, a comparison with this model suggests that the BTSP model provided by Milstein et al. could be consistent with a coordination of plasticity, as acknowledged by the authors at the end of the discussion.

We had cited Payeur et al., in the context of implications for artificial learning systems. We have now added a citation in the context of objective-based learning through mismatch of local circuit output and target information relayed by long-range feedback (P24, L634). We agree that comparing and contrasting these models in detail would be very interesting, but perhaps a more appropriate place for this would be in a follow-up review article.

The model is also partially related with the model of Graupner,.… Brunel. Recently used to capture STDP phenomena by Aljadeff… Debanne. There, two threshold are present, one for LTP and one for LTD. So similar to the α^+^ and α^-^ being different. But the dependence in that model is very limited.

We have now added the following citations in the context of the weight-dependent model and opposing plasticity processes (P16, L406):

Graupner M, Brunel N. STDP in a bistable synapse model based on CaMKII and associated signaling pathways. PLoS Comput Biol. Nov 2007;3(11):e221. doi:10.1371/journal.pcbi.0030221

Inglebert Y, Aljadeff J, Brunel N, Debanne D. Synaptic plasticity rules with physiological calcium levels. Proc Natl Acad Sci U S A. Dec 29 2020;117(52):33639-33648. doi:10.1073/pnas.2013663117

I thought the description of how W was kept between 1 and W_max_ was lacking. There are multiple ways of doing this. And the different types of implementation (hard threshold vs soft threshold) have been shown to lead to dramatically different functions in the STDP modelling literature.

Please see our response to the comments of Reviewer #3 below.

I think there is a typo in the list of parameters given at the end as I don't find τ_IS_ and τ_ET_, but only τ_I_ and τ_E_.

This oversight in the Materials and methods section has been fixed.

Reviewer #3 (Recommendations for the authors):This paper is beautifully done and well written. I have just a few suggestions:1) It seems a pity that after elegantly formulating a model of BTSP, the authors final formulation of the synaptic change in Equations 3 and 4 is ill-defined. This is because the relationship between W and W_init_ is not specified. Wouldn't a proper formulation state that W_init_ = W, Equation 3 does not describe dW/dt but rather a factor that determines ΔW through Equation 4, and that this latter process only takes place after a delay?Alternatively, could W_init_ in the equation for dW/dt just be set to W? This would have the nice property of keeping W bounded between 0 and W_max_. Does this not work?

Indeed, in the two-state finite resource model that we had in mind, W is updated continuously in real-time, and is naturally bounded between 0 and W_max_. In practice, for simplicity and efficiency of computation during parameter optimization, we updated W only once per lap. We agree with the Reviewer that presenting the model in the context of this numerical approximation diluted the message. We have revised the Results section accordingly and have removed all instances of W_init_. Figure 5 has been updated to show results obtained by updating W continuously. No change in parameters were necessary. We also regenerated Figure 6, keeping parameter values the same, but updating W continuously (Author response image 1). They are equivalent to those results obtained with the approximate method in the original Figure 6, which validates the original approach. As such, we have chosen not to regenerate or replace Figures 6, 7, Figure 6—figure supplements 1 and 2, or Figure 7—figure supplement 1.

**Author response image 1. sa2fig1:** 

We have also added the following text to the Materials and methods:P32, L862: “Changes in synaptic weight Wi were calculated by integrating the net rate of change of synaptic weight dWi/dt over the duration of plasticity induction. […] Equivalent results were obtained by updating Wi continuously in 10 ms steps without requiring any change in parameters.”